# Heterogeneous graph adaptive flow network

**Lu Yiqi**                                                        *yiqi001@e.ntu.edu.sg*
*Department of Electrical and Electronic Engineering*
*Nanyang Technological University*

**Ji Feng**                                                        *jifeng@ntu.edu.sg*
*Department of Electrical and Electronic Engineering*
*Nanyang Technological University*

**Tay Wee Peng**                                                  *wptay@ntu.edu.sg*
*Department of Electrical and Electronic Engineering*
*Nanyang Technological University*

**Reviewed on OpenReview:** *https://openreview.net/forum?id=usvg3yhjAx*

## Abstract

Many graphs or networks are heterogeneous by nature, involving various vertex types and relation types. Most graph learning models for heterogeneous graphs employ meta-paths to guide neighbor selections and extract composite relations. However, the use of meta-paths to generate relations between the same vertex types may result in directed edges and failure to fully utilize the other vertex or edge types in the data. To address such a limitation, we propose Heterogeneous graph adaptive flow network (HetaFlow), which removes the need for meta-paths. HetaFlow decomposes the heterogeneous graph into flows and performs convolution across heterogeneous vertex and edge types, using an adaptation to change the vertex features based on the corresponding vertex and edge types during aggregation. Experiments on real-world datasets for vertex clustering and vertex classification demonstrate that HetaFlow outperforms other benchmark models and achieves state-of-the-art performance on commonly used benchmark datasets. The codes are available at https://github.com/AnonymizedC/HetaFlow.

## 1   Introduction

A significant number of real-world graphs come with a diversity of vertex types and relation types. For example, the ACM dataset (Wang et al., 2019) (example data shown in Fig. 1) consists of four types of vertices: paper, author, conference, and subject. Many heterogeneous graph neural network (HGNN) models are based on the idea of meta-paths (Huang et al., 2016; Zhang et al., 2018a; Hu et al., 2018). A meta-path is an ordered sequence of edge types, which defines a new composite relation between the source vertex type and the destination vertex type. For example, in the ACM dataset, a relation between two papers can be described by the meta-path Paper-Author-Paper (P-A-P), which represents an author's co-authorship of these two papers; or Paper-Subject-Paper (P-S-P), which implies that the two papers investigate the same subject. Depending on the meta-path, the relation between vertices can have different semantics.

Most meta-path-based heterogeneous graph models for semi-supervised vertex classification and clustering (Zhang et al., 2019a; Shi et al., 2018; Fan et al., 2019) follow two similar steps to create homogeneous graphs on which traditional GNN models can then be applied: 1) Meta-paths are chosen so that the source and destination vertex types are the same. 2) Then, one employs a set of predefined meta-paths to decompose the original heterogeneous graph into one or several subgraphs with homogeneous vertex types. An appropriate GNN model on each subgraph can then be applied and the results fused to obtain the final inference.

People also proposed non-meta-path-based models. These models mainly follow two approaches. They either introduce some projection methods and attention mechanisms to aggregate representations of vertices, like HGT Hu et al. (2020) and HetSANN Hong et al. (2020), or find suitable meta-paths by themselves, like ie-HGCN Yang et al. (2021). These two directions have different challenges.

Methods of the first direction normally set different weights for different edge types. For example, HGT Hu et al. (2020) assigns different aggregation weights for each edge type. However, edges in heterogeneous graphs are very likely to be directed, and edges of one edge type may connect different vertex types. So, there is room to improve in the projection and aggregation parts, like introducing a type-related adjustment part during aggregation. Although the other kind of non-meta-path-based models can automatically find suitable meta-paths, they still need to face problems caused by the usage of meta-paths. For instance, the obvious direction of meta-path and the information loss of intermediate vertices.

- Most heterogeneous graph models fail to consider the directions of edges. In some heterogeneous graph applications, edges have an inherent direction based on the vertex types of their end vertices. For example, in a network connecting teachers and students, viewing an edge going from a teacher to a student versus the opposite direction has different meanings. Therefore, the importance of a vertex to another vertex depends on how that vertex is reached and not just on the distance or edge attention weights between them. Meta-path-based methods are more likely to meet such challenges, like the meta-path Subject-Paper-Conference-Subject (S-P-C-S) in Fig. 1. Thus, it may be appropriate to assign different weights to neighboring vertices according to their positions. Allocating the same weight to vertices at the same distance may result in suboptimal results.

- Many existing heterogeneous graph models failed to exploit the type information fully. It makes sense that features of different vertex types should be projected to different spaces, and aggregations along various edge types should use separate filters. Similarly, it is also appropriate to assign different weights to the same vertex when aggregating with vertices of different types (even when the edges are of the same type). Thus, the performance can be improved by introducing type-based adjustment weights during aggregations.

- As shown in Fig. 1(d), all the intermediate vertices and the vertices that do not appear in the chosen meta-paths are not represented in the homogeneous subgraphs created using the meta-path approach. Furthermore, all subgraphs only contain a single vertex type. A substantial proportion of the vertex features from vertex types not represented are discarded, which is apparently detrimental (e.g., HERec (Shi et al., 2018) and HAN (Wang et al., 2019)). Embedding methods such as MAGNN (Fu et al., 2020) employ encoders to encode the features of intermediate vertices. However, it is unclear how one can choose an efficient encoder for different datasets. Furthermore, these embeddings use type-based vertex feature transformations, which do not take the intermediate connections into consideration.

- The choice of meta-paths limits the relationships the model can learn. As shown in Fig. 1(c), the meta-path P-A-P allows a model to learn the relationship between two papers with a common author. It does not however learn other relationships like P-S-P if this meta-path has not been explicitly included.

- The number of possible meta-paths increases exponentially with respect to the number of vertex types and edge types. There is a positive correlation between the number of filters that are needed and the number of meta-paths. Most models in the literature Dong et al. (2017); Shi et al. (2018); Fu et al. (2020); Fu & King (2024) either discard some meta-paths or require significant computational power if all meta-paths are included. Meta-path-based approaches are thus not scalable if they want to include all possible meta-paths.

We introduce parallel flow decomposition and type-based adjustment to address these challenges. We decompose the graph into 1D paths which allows us to assign different weights to vertices based on their positions instead of distances.

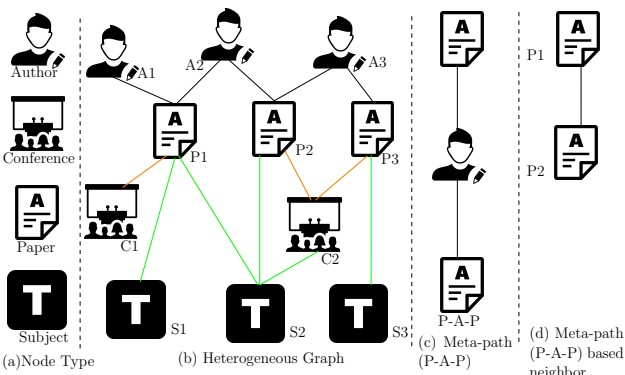

Figure 1: An example of a heterogeneous graph based on the ACM dataset.

We propose a novel Heterogeneous graph adaptive flow network (HetaFlow). HetaFlow first decomposes the graph into several groups of paths, which we call "parallel flows". Then, for each edge, the model encodes it according to its edge type and its two end vertex types. HetaFlow applies type-specific linear transformations to project heterogeneous vertex attributes for each vertex to a common vector space. Each edge thus gets its type-based edge feature. Next, HetaFlow performs intra-path aggregation along each path in every parallel flow, and inter-flow aggregations across multiple parallel flows with type-based feature adjustments.

The computational complexity of HetaFlow is lowered significantly compared to these other models and its model size depends only on the number of parallel flows used. See the discussion about model size and complexity in Appendix B and Appendix C.

In this work, we introduce the concepts of paths and parallel flows to heterogeneous graphs and design type-based adjustments. We generalize the model to heterogeneous graphs with additional mechanisms to handle the heterogeneity of the graphs. Extensive numerical experiments are provided to verify our module choices.

## 2 Related work

### 2.1 Graph neural networks

GNNs extend convolution neural networks (CNNs) to general graph-structured data. There are two main classes of GNNs, namely spectral-based and spatial-based. Spectral-based GNNs operate in the Fourier domain. An example is ChebNet (Defferrard et al., 2016), which processes graph signals (vertex features) using Chebyshev polynomials of the graph Laplacians. By simplifying the parameters of ChebNet, GCN (Kipf & Welling, 2016) overcomes the overfitting problem of ChebNet. However, spectral-based GNNs require the whole graph as an input in the worst case when the polynomial degrees are high, and are sensitive to graph structures. They thus suffer from scalability issues.

Spatial-based GNNs imitate the convolution operation in CNNs. GNNs of this kind operate directly on a vertex's neighbors. An example is GraphSAGE (Hamilton et al., 2017), which learns embeddings by aggregating features of the target vertex and its neighbor vertices. Other spatial-based GNN variants have been proposed by improving the aggregator function. GAT (Velickovic et al., 2018) employs the attention mechanism in the aggregator to assign different importance weights based on the neighboring vertices' features. GFCN (Ji et al., 2020) decomposes a graph into non-intersecting paths and performs convolutions along these paths before aggregating them. It can be shown that GFCN recovers the GCN model. Inspired by the idea of GGNN (Li et al., 2016) that adds gated recurrent unit (GRU) (Cho et al., 2014) into the aggregator function, GaAN (Zhang et al., 2018b) combines GAT and GGNN for spatio-temporal graphs. To improve the accuracy, STAR-GCN (Zhang et al., 2019c) employs various GCN encoder-decoders.

The above-mentioned GNNs are designed for homogeneous graphs in which all vertices and edges are of the same type. One main limitation that stops GNNs from being adapted to heterogeneous graphs is that the vertex features for different vertex types in a heterogeneous graph are from different spaces.

Adaptive graph learning (Luo et al., 2017; Li et al., 2018; Zhou et al., 2019) has shown good generalization ability when facing real problems, where data are on an irregular grid or in other kinds of non-Euclidean domains. Adaptive graph learning can help models to be more general. To obtain a discriminative feature subset, Luo et al. (2017) learns the optimal reconstruction graph and selective matrix simultaneously. The paper Zhou et al. (2019) combines multi-feature dictionary learning and adaptive multi-feature graph learning into a unified learning model. To parameterize the similarity between two vertices on the graph, Li et al. (2018) employs a learnable distance metric. In this work, we design a type-based adaptive method to parameterize the filters in some layers so that our model can better handle heterogeneous graphs.

## 2.2 Heterogeneous graph embedding

Various heterogeneous graph embedding methods have been proposed to embed vertex features into a low-dimensional space. For example, metapath2vec (Dong et al., 2017) employs meta-path-based random walks and skip-gram (Mikolov et al., 2013a) to generate vertex embeddings. With the guidance of manually-defined meta-paths, ESim (Shang et al., 2016) learns vertex features in a user-preferred embedding space. HIN2vec (Fu et al., 2017) learns vertex embeddings and meta-path representations simultaneously by carrying out multiple prediction training tasks. Given a meta-path, HERec (Shi et al., 2018) proposes an algorithm that converts a heterogeneous graph into meta-path-based neighbors and then employs the DeepWalk (Perozzi et al., 2014) model to learn the target vertex embedding. Similar to HERec, HAN (Wang et al., 2019) also decomposes a heterogeneous graph into multiple meta-path-based homogeneous graphs. HAN uses a graph attention mechanism to combine results from different meta-paths.

These heterogeneous graph embedding models suffer from the limitations discussed in Section 1 due to the reliance on meta-paths to construct homogeneous subgraphs. The use of meta-paths inevitably introduces some limitations like discarding vertex features or loss of vertices and connections. For example, metapath2vec, ESim, HIN2vec, and HERec omit vertex features, which lowers their performance when performing tasks on graphs with rich vertex features. Furthermore, HERec and HAN only consider the two end vertices of each meta-path while ignoring the vertices along the path, which results in information loss. The method metapath2vec achieves suboptimal performance as it takes only one meta-path as the input. SeHGNN(Yang et al., 2023) improves the performance by introducing feature projection. However, it needs a complicated extra data pre-processing step.

Several HGNNs (Zhang et al., 2019b; Yun et al., 2019; Lv et al., 2021; Ahn et al., 2022) have been proposed to solve the meta-path selection dilemma by generating new graph structures. For instance, HetGNN (Zhang et al., 2019b) divides neighbors into subsets based on their types and employs an aggregator function for each type of neighbor. However, as pointed out by Lv et al. (2021), HetGNN has "information missing in homogeneous baselines" and "GAT with correct inputs gets clearly better performance". Simple-HGN (Lv et al., 2021) is based on GAT. It leverages type information by introducing learnable type embeddings and enhanced modeling power with residual connections and $l_2$ normalization on the output embeddings.

Many attention-based models failed to learn node-level and relation-level attention simultaneously. BA-GNN Iyer et al. (2021) employs a novel bi-level graph attention mechanism to overcome this issue. Instead of from the global graph context, BA-GNN attends to both types of information from local neighborhood contexts. It achieved good performance. However, it assigns the same weights to the neighbor vertices of the same distance, which means the directions of edges are not exploited during aggregation.

ie-HGCN Yang et al. (2021) proposed a hierarchical architecture as object-level and type-level aggregation. ie-HGCN can learn the representations of objects and automatically discover and exploit the most useful meta-paths. Its 'two-level aggregation structure' shows good performance and interpretability in capturing semantic relations. Our HetaFlow has a similar design. HetaFlow's intra-path aggregation and inter-path aggregation empower it to catch semantic relations efficiently. As mentioned in Section 1, ie-HGCN fails to consider the directions of edges. It assigns the same weights to neighbor vertices in the same distance. By

Table 1: Notations

| Notations | Description |
|---|---|
| $\mathbb{R}^n$ | The $n$-dimensional Euclidean space. |
| $\mathcal{G} = (\mathcal{V}, \mathcal{E})$ | Graph with vertex set $\mathcal{V}$ and edge set $\mathcal{E}$. |
| $\mathcal{A}$ | The set of vertex types. |
| $\mathcal{R}$ | The set of edge types. |
| $T_u$, $T_{uv}$ | The embedding of a vertex $u$ and edge $(u, v)$, respectively. |
| $\mathcal{P}_v$ | The collection of paths that contain the vertex $v$. |
| $\mathcal{N}_v(h)$ | The $h$-hop neighborhood of vertex $v$. |
| $\mathbf{x}_v$ | Feature vector of vertex $v$. |
| $\mathbf{h}_v$ | Hidden state (embedding) of vertex $v$. |
| $\sigma(\cdot)$ | Activation function. |
| $\odot$ | Element-wise multiplication. |
| $\lvert \cdot \rvert$ | The cardinality of a set. |
| $\parallel$ | Vector concatenation. |

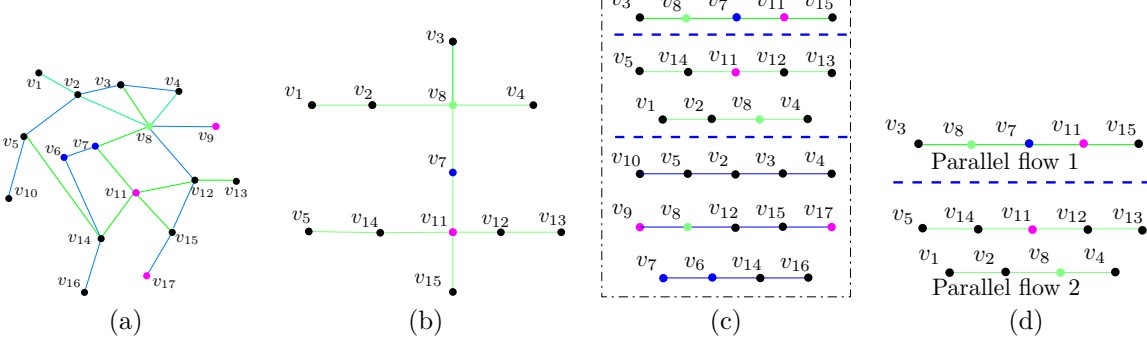

Figure 2: (a) A heterogeneous graph: different colors for vertices and edges indicate different types. (b) A subgraph with homogeneous edge type. (c) A parallel flow decomposition. (d) Two parallel flows.

decomposing the graph into 1-D flows, HetaFlow is capable of learning different weights to neighbors with the same distance.

HGT Hu et al. (2020) and HetSANN Hong et al. (2020) are also non-meta-path-based methods. They avoid the meta-path selection dilemma by projecting the representations to the same space and employing attention mechanisms to help aggregations. They make use of the 'type' information by assigning different aggregation weights to various edge types. However, this may not be enough since the same-type edges may connect vertices of different vertex types, which usually happens when there exists only one edge type in the heterogeneous graph. HetaFlow mitigates this problem by introducing an adjustment component. HetaFlow employs learnable parameters to generate adjustment weights based on the embeddings of types of vertices and edges. When aggregating on the same edge type, the features of neighbor vertices are adjusted differently based on their vertex types.

## 3 Preliminaries

In this section, we define important terminologies related to heterogeneous graphs. Graphical illustrations are provided in Fig. 2. We summarize the commonly used notations in this paper in Table 1. Throughout this paper, we consider a graph $\mathcal{G} = (\mathcal{V}, \mathcal{E})$, where $\mathcal{V}$ is the set of vertices and $\mathcal{E}$ is the set of edges of the graph.

**Definition 1.** *(Heterogeneous graph) A heterogeneous graph $\mathcal{G}$ is associated with a vertex type mapping function $\phi : \mathcal{V} \mapsto \mathcal{A}$ and an edge type mapping function $\psi : \mathcal{E} \mapsto \mathcal{R}$, where $\mathcal{A}$ and $\mathcal{R}$ denote the predefined sets of vertex types and edge types, respectively, with $|\mathcal{A}| + |\mathcal{R}| > 2$. Furthermore, if a subgraph $\mathcal{G}'$ includes only a single edge type, then we call $\mathcal{G}'$ a subgraph of $\mathcal{G}$ with homogeneous edge type.*

An example of a heterogeneous graph is shown in Fig. 2(a). This heterogeneous graph consists of four different types of vertices and two types of edges. Fig. 2(b) illustrates a subgraph with homogeneous edge type. Though it contains only the green edge type, there are four types of vertices in it, which means it is also a heterogeneous graph.

**Definition 2.** *(Path) A path $P$ of a heterogeneous graph $\mathcal{G}$ is a connected subgraph such that each vertex of $P$ has a degree not more than 2 in $P$. In particular, a single vertex or a cycle is a path.*

**Definition 3.** *(Parallel flow) Two paths $P_1$ and $P_2$ are said to be parallel to each other if they do not share any common vertices in $\mathcal{G}$. Furthermore, a set of paths $\mathcal{P} = \{P_1, \ldots, P_n\}$ is a parallel flow if its elements are pairwise parallel.*

For example, in Fig. 2(a), consider the path $P_1$ given by the vertex sequence $(v_{10}, v_5, v_2, v_3, v_4)$, the path $P_2 = (v_{16}, v_{14}, v_6, v_7, v_8, v_9)$ and the path $P_3 = (v_{17}, v_{15}, v_{12}, v_{13})$, we can see any two of $P_1$, $P_2$, and $P_3$ are parallel since there are no common vertices or edges between them. Furthermore, the set $\{P_1, P_2, P_3\}$ is a parallel flow as all its elements are pairwise parallel.

**Definition 4.** *(Parallel flow decomposition) Consider a collection of parallel flows $\mathcal{P}_1, \ldots, \mathcal{P}_m$, each of which contains paths of homogeneous edge type. If $\bigcup_{i=1}^{m} \mathcal{P}_i$ contains all edges in the heterogeneous graph $\mathcal{G}$, then this collection of parallel flows is called a parallel flow decomposition of $\mathcal{G}$.*

Since each edge in Fig. 2(b) appears in Fig. 2(d) and the two parallel flows do not share common edges, Fig. 2(d) is an example of a parallel flow decomposition of Fig. 2(b). Parallel flow decompositions are not unique for a heterogeneous graph $\mathcal{G}$. For instance, in Fig. 2(b), we can choose a parallel flow consisting of a single path $v_3$-$v_8$-$v_7$-$v_{11}$-$v_{12}$-$v_{13}$, and a parallel flow consisting of two four-vertex paths $v_1$-$v_2$-$v_8$-$v_4$ and $v_5$-$v_{14}$-$v_{11}$-$v_{15}$. This forms a parallel flow decomposition of Fig. 2(b) different from that in Fig. 2(d).

Besides accuracy, runtime speed or computational complexity of a GNN is a concern for large graphs. The computational complexity is closely related to the model size. To quantify our model size, we require the following definitions and upper bound on the number of parallel flows (Ji et al., 2020). We briefly recall the following notions and results from Ji et al. (2020) for the homogeneous graphs.

**Definition 5.** *A set of parallel flows $\{\mathcal{P}_1, \ldots, \mathcal{P}_m\}$ is a cover of $\mathcal{G}$ if the union of all the paths in the parallel flows contains all the edges of $\mathcal{G}$. The smallest $m$ such that there is a cover consisting of $m$ parallel flows is denoted as $\mu(\mathcal{G})$.*

## 4 Heterogeneous adaptive flows

In this section, we present HetaFlow. We refer the reader to Fig. 3 for an illustration of the HetaFlow framework, which is summarized as follows:

- We first explain the motivation for introducing such decomposition methods. See Section 4.1.

- We perform a parallel flow decomposition of a given heterogeneous graph $\mathcal{G}$, with each parallel flow having a single edge type. See Section 4.2.

- We perform vertex feature transformation to map the features of all vertices to the same space. See Section 4.3.

- Intra-path aggregation with attention weights is performed along each path in each parallel flow. See Section 4.4.

- A vertex may appear in multiple parallel flows, each of which generates a feature vector in the steps above. Inter-flow aggregation is performed to fuse these feature vectors for each vertex. See Section 4.5.

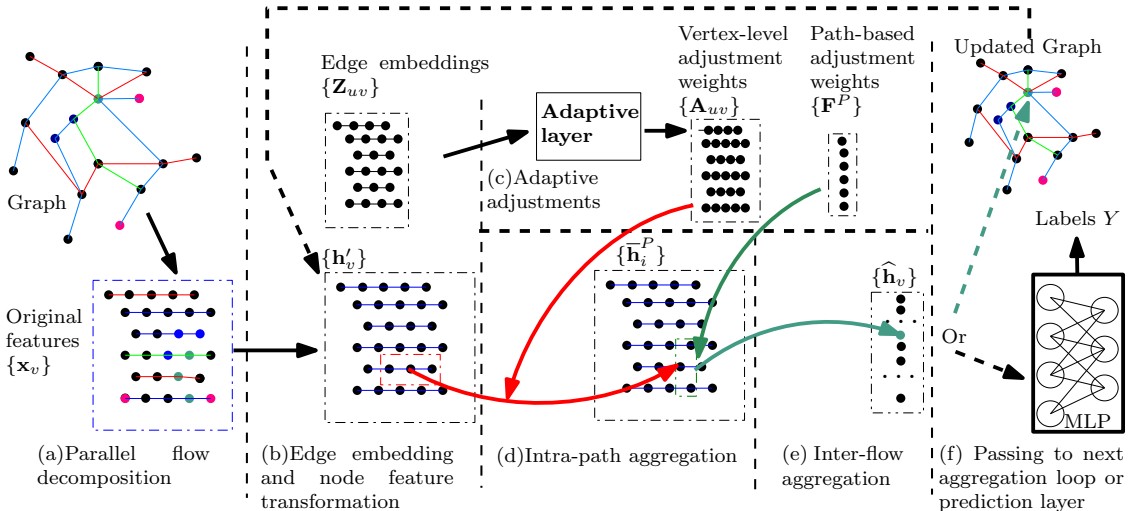

Figure 3: The overall framework of HetaFlow. Steps (c), (d), and (e) can be repeated.

## 4.1 Motivation and differences

Before delving into the details of the model, we first discuss why we choose to decompose the heterogeneous graph into parallel flows. Without relying on any prior meta-path knowledge, we want to decompose a heterogeneous graph into subgraphs that traditional GNNs can handle. An intuitive idea is to form subgraphs by selecting the edges that follow similar patterns, i.e., build subgraphs with homogeneous edge type. For example, consider the heterogeneous graph shown in Fig. 2(a). It can be decomposed into two subgraphs: one subgraph is shown in Fig. 2(b) and the other subgraph consists of the three separate blue lines in Fig. 2(a). However, even though a subgraph has homogeneous edge types, it may still not be appropriate to apply a homogeneous GNN on the subgraph due to the following two considerations.

Firstly, there may exist multiple vertex types in each subgraph, and thus we need to make adjustments based on the types of the vertices when performing convolutions. We resolve this challenge by introducing vertex feature transformation, which is discussed in detail in Section 4.3. As discussed in the comparisons in Section 2.2, many non-meta-path-based methods, like HGT Hu et al. (2020) and HetSANN Hong et al. (2020) employ different weights for the aggregations along each edge type. However, edges of one edge type may connect various types of vertices. For instance, 'paper' type vertices can be linked to both 'conference' type and 'journal' type vertices by 'be submitted to' type edge. When predicting the research area of a 'paper' vertex, it is better to assign different weights to 'conference' type and 'journal' type vertices because journals are more general in some research area. However, HGT and HetSANN shall use the same filter since the edge type is the same. We introduce the adaptation adjustments to improve the performance against this problem.

Secondly, some types of edges may be directed, like the "teacher$_i$-teaches-student$_j$" relations in a social network. Consider a path with 3 vertices and two edges, "$v_i$-teaches-$v_j$-teaches-$v_k$", where each vertex stands for a person. When trying to predict the academic interest of $v_j$, we may find features of $v_i$ and $v_k$ are of different importance. Thus, it may be appropriate to assign different weights to neighboring vertices according to their positions instead of distances. Allocating the same weight to vertices at the same distance may result in suboptimal results as the weights no longer capture the importance of these vertices to the target vertex at which the convolution is applied.

We introduce parallel flow decomposition that decomposes the graph into 1D paths and allows us to assign different weights to vertices based on their positions in a neighborhood of the target vertex on the 1D path. Moreover, though the edges of each subgraph are likely to follow similar patterns, they may contain different meanings and need to be processed differently. For example, consider a computer vision task. In an image, the horizontal and vertical edges follow the same pattern (both can be detected by similar 1D convolution

filters) but may have different meanings and contributions when performing prediction tasks. Parallel flow decomposition helps us to capture such information by splitting the graph into different flows.

However, there's a certain degree of randomness to this decomposition process. Parallel flow decomposition was originally designed for homogeneous graphs, which means it failed to consider the relations between the flows of different subgraphs. Note that 'type information' is now available, as well as some semantic relations between the flows of various subgraphs. HetaFlow designs a transformer-based semantic adaptation module and employs an attention mechanism (step (c)(d)(e) in Fig. 3) to learn the mutual attention between semantic flows (see in Section 4.3, Section 4.4, Section 4.5). The whole forward propagation process is illustrated in Algorithm 1.

## 4.2   Flow decomposition

### 4.2.1   Implementation

Let $\mathcal{O}$ be an operation that takes a heterogeneous graph $\mathcal{G}_0 = (\mathcal{V}, \mathcal{E}_0)$ as the input and outputs a subgraph $\mathcal{G}_1 = (\mathcal{V}_1, \mathcal{E}_1)$ with homogeneous edge type. One intuitive example of $\mathcal{O}$ is to select all edges of a certain type in $\mathcal{G}_0$. Starting with a given heterogeneous graph $\mathcal{G}_0$, we build a subgraph $\mathcal{G}_1 = \mathcal{O}(\mathcal{G}_0)$. Then, we repeat this operation on the remaining part, i.e., the complement of graph $\mathcal{G}_1$ with respect to $\mathcal{G}_0$, to obtain another homogeneous edge type subgraph $\mathcal{G}_2 = \mathcal{O}(\mathcal{G}_1^{\mathsf{c}})$, where $\mathcal{G}_1^{\mathsf{c}} = (\mathcal{V}, \mathcal{E}_0 \backslash \mathcal{E}_1)$. This procedure is repeated until no edges remain.

The original heterogeneous graph is thus decomposed into several subgraphs with homogeneous edge types. Then, we can follow Ji et al. (2020) to decompose each subgraph into parallel flows and take the union of the decompositions. This union is a parallel flow decomposition of the original heterogeneous graph.

For completeness, we now provide a brief description of the parallel flow decomposition implementation. Note that this approach is only one of many possible approaches. For details, we refer the reader to Ji et al. (2020). For every subgraph $\mathcal{G}_i = (\mathcal{V}_i, \mathcal{E}_i)$ (can be possibly disconnected), a forest $\mathcal{G}_{i1} = (\mathcal{V}_{i1}, \mathcal{E}_{i1})$ consisting of trees that span the components of $\mathcal{G}_i$ is found by performing a depth-first search of the graph (in this paper, we select a vertex with the highest degree in subgraph $\mathcal{G}_i$ to be the root vertex for each component). Denote the complement of the forest $\mathcal{G}_{i1}$ with respect to $\mathcal{G}_i$ as $\mathcal{G}_{i1}^{\mathsf{c}}$, whose edge set is $\mathcal{E}_i \backslash \mathcal{E}_{i1}$ and isolated vertices after removing $\mathcal{E}_{i1}$ are removed. Repeating the process, we again find a spanning forest $\mathcal{G}_{i2}$ for $\mathcal{G}_{i1}^{\mathsf{c}}$. This procedure is repeated until the remaining complement is empty. The procedure terminates in a finite number of steps $n$ because the number of edges remaining in the complement at each step decreases. We then have $\mathcal{G}_i = \bigcup_{j=1}^{n} \mathcal{G}_{ij}$.

Next, we consider the parallel flow decomposition of a tree in each forest $\mathcal{G}_{ij}$ obtained in the above procedure. Let this tree be $T$ and $\deg_{\max}$ be its maximal degree. We can label each edge in $T$ with a number in the set $\{1, 2, \ldots, \lfloor (\deg_{\max} + 1)/2 \rfloor\}$ such that every label is used at most twice for edges in $T$ connected to each vertex $v$ in $T$. Finally, to form a parallel flow, we take edges that have the same label. The tree $T$ is now decomposed into multiple parallel flows.

## 4.3   Feature embedding

We first record the types of vertices and edges so that we can adjust the vertex features according to their types. For most datasets, the vertex types and edge types are categorical. We use one-hot encoding for them.

For an edge $(u, v)$ connecting vertices $u$ and $v$, let $T_u \in \mathbb{R}^{|\mathcal{A}|}$ denote the one-hot encoding of the type of vertex $u$, and $T_{uv} \in \mathbb{R}^{|\mathcal{R}|}$ be the one-hot encoding of the type of edge $(u, v)$. Then, its edge embedding is given by:

$$\mathbf{Z}_{uv} = \mathbf{T}_u \parallel \mathbf{T}_{uv} \parallel \mathbf{T}_v. \tag{1}$$

Different vertex types may have feature vectors with unequal dimensions and belong to different feature spaces. So we apply a type-based linear transformation for each type of vertices before feeding vertex feature vectors into the intra-path aggregation module. Let $\mathbf{x}_v \in \mathbb{R}^{d_A}$ denote the original feature vector of vertex $v$, and $\mathbf{W}_A \in \mathbb{R}^{d' \times d_A}$ denote the parametric weight matrix for type $A$ vertices. Then, for a vertex $v$ of type

$A \in \mathcal{A}$, the projected latent vector is given by

$$\mathbf{h}'_v = \mathbf{W}_A \mathbf{x}_v \in \mathbb{R}^{d'}. \tag{2}$$

### 4.4 Intra-path aggregation

Since the intra-path aggregation involves only vertices on this path, it is natural to consider 1D convolution. For each parallel flow $\mathcal{P}$, we employ the same 1D convolution filter of odd length $h$ for all paths $P \in \mathcal{P}$. Let $\mathcal{N}_v^P = P \cap \mathcal{N}_v((h-1)/2)$ to denote neighbors of vertex $v$ on path $P$. The 1D convolution at each vertex $v$ along the path $P$ is given by:

$$\widehat{\mathbf{h}}_v^P = \sigma \left( \sum_{u \in \mathcal{N}_v^P} w_u^{\mathcal{P}} \mathbf{h}'_u \right), \tag{3}$$

where the weights $w_u^{\mathcal{P}}$ are the 1D convolution filter weights for the parallel flow $\mathcal{P}$.

To fuse features learned from different edge types and parallel flows, we perform vertex-level and flow-based adjustments. The weighting coefficients employed in this process are called adjustment weights.

We design an adjustment weight that modifies the edge weights based on the types of edges and their end vertices. Let $\mathbf{W}_a$ and $\mathbf{b}_a$ be learnable parameters. Then, the vertex-level adjustment weight $\mathbf{A}_{uv}$ is defined as:

$$\mathbf{A}_{uv} = \sigma \left( \mathbf{W}_a \mathbf{Z}_{uv} + \mathbf{b}_a \right) \in \mathbb{R}^{d'}. \tag{4}$$

Suppose that vertex $u$ is a neighbor of vertex $v$ on path $P$. We modify (3) as:

$$\widehat{\mathbf{h}}_v^P = \sum_{u \in \mathcal{N}_v^P} \sigma \left( w_u^{\mathcal{P}} (\mathbf{h}'_u \odot \mathbf{A}_{uv}) \right). \tag{5}$$

Next, we use attention mechanisms to learn the importance $e_{vu}^P$ of a neighboring vertex $u$ to vertex $v$ on the path $P$, which is then normalized across all choices of $u \in \mathcal{N}_v^P$ using the softmax function. Then, we obtain a normalized importance weight $\alpha_{vu}^P$ for each neighboring vertex on this path.

Furthermore, to reduce the variance introduced by the parallel flow decomposition, we extend the attention mechanism to multiple heads by concatenating $K$ independent attention mechanisms. Let $\overline{\mathbf{h}}_i^P \in \mathbb{R}^{d'}$ denote the features of vertex $i$ after intra-path aggregation, $\left[ e_{vu}^P \right]_k$ denote the importance of $u$ to $v$ in the $k$-th head, $\left[ \alpha_{vu}^P \right]_k$ denote the normalized importance of $u$ to $v$ in the $k$-th head, and $[\mathbf{a}_P]_k \in \mathbb{R}^{2d'}$ the parameterized attention vector for path $P$ in the $k$-th head. Then, the $k$-th attention mechanism, for $k = 1, \ldots, K$, can be expressed as:

$$
\begin{aligned}
\left[ e_{vu}^P \right]_k &= \text{LeakyReLU} \left( \left[ \mathbf{a}_P^\top \right]_k \cdot \left[ \widehat{\mathbf{h}}_v^P \| \widehat{\mathbf{h}}_u^P \right] \right), \\
\left[ \alpha_{vu}^P \right]_k &= \frac{\exp \left( \left[ e_{vu}^P \right]_k \right)}{\sum_{s \in N_v^P} \exp \left( \left[ e_{vs}^P \right]_k \right)}, \\
\overline{\mathbf{h}}_i^P &= \Big\Vert_{k=1}^K \sigma \left( \sum_{j \in \mathcal{N}_i^P} \left[ \alpha_{ij}^P \right]_k \cdot \widehat{\mathbf{h}}_j^P \right).
\end{aligned} \tag{6}
$$

To fuse the features $\overline{\mathbf{h}}_v^P$ in (6) of the same vertex $v$ on different paths $P$, we incorporate an attention mechanism based on paths. Each path from a parallel flow consists of only one edge type. Let $T_P$ be the edge type encoding of path $P$, and $\mathbf{W}_f$, $\mathbf{b}_f$ be learnable parameters. The path-based adjustment weight $\mathbf{F}^P$ for path $P$ is defined as:

$$\mathbf{F}^P = \sigma \left( \mathbf{W}_f \mathbf{T}_P + \mathbf{b}_f \right) \in \mathbb{R}^{\overline{d}}. \tag{7}$$

We update the vertex features by doing an element-wise multiplication. The feature $\mathbf{h}_v^P \in \mathbb{R}^{d'}$ of vertex $v$ on path $P$, which is processed by the inter-flow aggregation in the next step, can be expressed as:

$$\mathbf{h}_v^P = \overline{\mathbf{h}}_v^P \odot \mathbf{F}^P. \tag{8}$$

### 4.5 Inter-flow aggregation

Suppose there are $M$ parallel flows in the parallel flow decomposition of Section 4.2. Then for each vertex $v$, since it belongs to at most one path in each parallel flow, we have at most $M$ latent vectors $\{\mathbf{h}_v^P : P \in \mathcal{P}_v\}$, where $\mathcal{P}_v$ is the collection of all paths that contain vertex $v$ and $|\mathcal{P}_v| \leq M$. Since paths in $\mathcal{P}_v$ are not equally important to the vertex $v$, it is reasonable to use the attention mechanism to assign different weights to features obtained from different paths. We use $\beta_v^P$ to denote the relative importance of path $P$ to vertex $v$. We perform a weighted sum of all the path-based feature vectors of $v$ using the parameterized attention vector $\mathbf{q} \in \mathbb{R}^{d'}$ as follows:

$$
\begin{aligned}
e_v^P &= \mathbf{q}^\top \mathbf{h}_v^P, \\
\beta_v^P &= \frac{\exp\left(e_v^P\right)}{\sum_{P \in \mathcal{P}_v} \exp\left(e_v^P\right)}, \\
\widehat{\mathbf{h}}_v &= \sum_{P \in \mathcal{P}_v} \beta_v^P \cdot \mathbf{h}_v^P.
\end{aligned}
\tag{9}
$$

Finally, we add a layer with a learnable weight matrix $\mathbf{W}_o \in \mathbb{R}^{d_o \times d'}$ to change the shape of the output vector:

$$\mathbf{h}_v = \sigma\left(\mathbf{W}_o \widehat{\mathbf{h}}_v\right). \tag{10}$$

### 4.6 Training

To process the vertex representations, we use the following loss functions. Here we denote the set of vertices that have labels as $\mathcal{V}_L$, the number of classes as $C$, the one-hot label vector of vertex $v$ as $\mathbf{y}_v$, and the predicted probability vector of vertex $v$ as $\mathbf{h}_v$. For semi-supervised learning, we use the cross entropy loss as follows:

$$\mathcal{L} = -\sum_{v \in \mathcal{V}_L} \sum_{c=1}^{C} \mathbf{y}_v[c] \cdot \log \mathbf{h}_v[c], \tag{11}$$

where $y_v[c]$ denotes the $c$-th component of the vector $y_v$.

For unsupervised learning, let $\Omega$ be the set of positive (observed) vertex pairs, and $\Omega^-$ be the set of negative vertex pairs sampled from the complement of $\Omega$. Then, we minimize the following loss function via negative sampling (c.f. Mikolov et al. (2013b)) with sigmoid function $\sigma(\cdot)$:

$$\mathcal{L} = -\sum_{(u,v) \in \Omega} \log \sigma\left(\mathbf{h}_u^\top \mathbf{h}_v\right) - \sum_{(i,j) \in \Omega^-} \log \sigma\left(-\mathbf{h}_i^\top \mathbf{h}_j\right). \tag{12}$$

### 4.7 Analysis of HetaFlow

#### 4.7.1 Generality

In this section we want to discuss about the generality of HetaFlow. It can represent more general polynomials than most current meta-path-based heterogeneous GNNs. It is even capable of achieving aggregations that can not be expressed by polynomials.

Assume that a given heterogeneous graph has $n$ types of edges and let $S$ denote a fixed graph shift operator (one common choice is the graph's normalized adjacency matrix or Laplacian matrix). For edge type $i$, we can pick out all corresponding elements in $S$ and perform zero paddings to form a submatrix $S_i$, whose shape is the same as $S$. In this way, we decompose $S$ into the sum of $n$ submatrices $S_1, \ldots, S_n$, where $S_i$ is the corresponding graph shift operator for edges of type $i$.

Many current heterogeneous GNNs perform convolutions among each subgraph and then fuse the outputs of each subgraph (Wang et al., 2019; Fu et al., 2020; Fu & King, 2024). For a given graph signal $X$, assume that a meta-path-based model fuses with the sum function. Then the final output is

$$\sigma(\sum_{i=1}^{n} p_i(S_i)X), \tag{13}$$

where $\sigma(\cdot)$ is the activation function, and $p_i(\cdot)$ denotes a given graph convolution filter for edge type $i$.

HetaFlow is capable of representing a more general polynomial, where each monomial consists of several variates. This observation is expressed in the following result.

**Proposition 1.** *Suppose no edge is contained in different paths inside a given set of parallel flows. If $M$ is the number of monomials and $N$ denotes the largest number of variates contained in one monomial, then there is a HetaFlow model with $M + 2\sum_{i=1}^{M}\sum_{j=1}^{N} k_{ij}$ hidden layers producing the same output as the convolution filter*

$$p(S_1, \ldots, S_n) = \sum_{i=1}^{M} b_i \prod_{j=1}^{N} S_{a_{ij}}^{k_{ij}} \tag{14}$$

*for any input graph signal, where $a_{ij}$ denotes the index of the $j$-th variate in the $i$-th monomial, and $k_{ij}$ and $b_i$ denote the corresponding power and coefficient, respectively.*

*Proof.* See Appendix A. $\square$

Proposition 1 shows that there is no loss in generality in adopting HetaFlow with parallel flow decomposition when compared to traditional heterogeneous GNNs like HAN Wang et al. (2019), MAGNNFu et al. (2020), and Simple-HGN(Lv et al., 2021). It is clear from (14) that the representations we can learn using HetaFlow are *strictly* richer than that achievable by the heterogeneous GNNs in (13). This motivates us to consider the use of parallel flow decompositions in HetaFlow.

In contrast, (14) is a multi-variate polynomial and involves *interactions* between different edge types with the same weights assigned to the same kind of interaction. It is also clear that HetaFlow is not restricted to that particular form. By utilizing 1D convolutions along *paths*, HetaFlow can assign *different* weights to vertices at the same distance in the path (i.e., the 1D filter is non-symmetric). And under this case, it is obvious that we can not use any polynomial to represent it.

In Section 5.4, we perform ablation studies on the effect of introducing parallel flow decompositions to obtain insights into the impact of different decomposition approaches.

### 4.7.2 Time complexity

We provide a detailed general analysis of the model size and computational complexity of HetaFlow in Appendix B and Appendix C. Here we follow the assumptions of SeHGNN(Yang et al., 2023) to make clearer comparisons with other models. The theoretical results are summarized as Table 2

We assume a one-layer structure with $M$ meta-paths for SeHGNN and HAN. Let $l$ be the maximum hop of meta-paths. To ensure the same receptive field size, we assume a $l$-layer structure for Simple-HGN and HetaFlow. The dimension of input and hidden vectors is $d$. Instead of considering all vertices, here we discuss the case when doing tasks on vertices of one vertex type. The number of target-type vertices is $n$. For HAN, let $e_1$ be the average number of neighbors in meta-path neighbor graphs. Let $e_2$ be involved neighbors during

multi-layer aggregation on Simple-HGN. Both $e_1$ and $e_2$ grow exponentially with the length of metapaths and layer number $l$. (19) shows that the computational complexity of HetaFlow is closely related to 'the number of edge types' and 'the max degree of vertices'. Let $|\mathcal{R}'|$ denotes the total number of edge types of the involved subgraphs, and $|\mathcal{P}'|$ denotes the total number of parallel flows to be aggregated.

Then from (19), the time complexity of HetaFlow is:

$$\mathcal{O}(|\mathcal{R}'||\mathcal{P}'|nd^2 + nld^2) = \mathcal{O}((|\mathcal{R}'||\mathcal{P}'| + l)nd^2). \tag{15}$$

For complicated graphs whose number of types of vertices and edges are large, $|\mathcal{R}'|$ is normally much smaller than the total number of edge types $|\mathcal{R}|$. There are $|\mathcal{A}|^2$ possible edge types and edges of at most $2|\mathcal{A}| - 1$ edge types connecting target-type vertices (for the generality of discussion, we consider the directed graph here. If it is undirected, then $|\mathcal{R}'|$ is at most $|\mathcal{A}|$). So, HetaFlow only needs to decompose at most $2|\mathcal{A}| - 1$ subgraphs. Moreover, HetaFlow can discard parallel flows that do not contain target-type vertices. For instance, consider the example claimed in Fig. 2, HetaFlow can discard parallel flow 2 in Fig. 2(d) if the target vertex type is the blue vertex type. So $|\mathcal{P}'|$ is normally smaller than the total number of parallel flows $|\mathcal{P}|$. And the number of possible meta-paths is large for complicated graphs. Take length 5 meta-paths as an example (we assume the endpoints of meta-paths are of the target node type), each intermediate position on the meta-path has $|\mathcal{A}|$ possibilities. There are $|\mathcal{A}|^3$ possible meta-paths that contain target vertices as the endpoints. Thus, $M \gg |\mathcal{R}'||\mathcal{P}'|$ when trying to consider all possible meta-paths for complicated graphs.

For the four datasets we tested in experiments, if want to consider all possible meta-path, then we have $M \gg |\mathcal{R}'||\mathcal{P}'|$. so the theoretical complexity of HetaFlow is much lower than that of SeHGNN, and much lower than that of HAN and Simple-HGN according to (Yang et al., 2023).

| | Feature projection | Neighbor aggregation | Semantic fusion | Total |
|---|---|---|---|---|
| SeHGNN | $\mathcal{O}(nMd^2)$ | – | $\mathcal{O}(n(Md^2+M^2d))$ | $\mathcal{O}(nd(M^2+Md))$ |
| HAN | $\mathcal{O}(nd^2)$ | $\mathcal{O}(nMe_1d)$ | $\mathcal{O}(nMd^2)$ | $\mathcal{O}(nd(Me_1+Md))$ |
| Simple-HGN | $\mathcal{O}(nld^2)$ | $\mathcal{O}(ne_2d)$ | | $\mathcal{O}(nd(e_2+ld))$ |
| HetaFlow | $\mathcal{O}(nld^2)$ | $\mathcal{O}(|\mathcal{R}'||\mathcal{P}'|nd^2)$ | | $\mathcal{O}((|\mathcal{R}'||\mathcal{P}'|+l)nd^2)$ |

Table 2: Time complexity of HetaFlow, SeHGNN, HAN, and Simple-HGN.

# 5 Experiments

In this section, we conduct experiments on the clustering and classification tasks using benchmark datasets. We compare the performance of HetaFlow to other state-of-the-art baseline models. The implementation details are claimed in Appendix D. After testing with benchmarks, we further conduct experiments on several variants of HetaFlow to validate the effectiveness of each component of our model. We test the validity of vertex feature adjustment, the importance of introducing parallel flow decomposition, and the influence of different decomposition methods separately. These ablation studies are illustrated in Appendix E.

## 5.1 Datasets

We test the performance of HetaFlow on the Heterogeneous Graph Benchmark (HGB) (Lv et al., 2021), using the following vertex classification datasets:

- **ACM**. We choose the papers published in SIGMOD, KDD, MobiCOMM, SIGCOMM, and VLDB. The papers belong to three categories: Data Mining, Wireless Communication, and Database. Then we construct a subset that comprises 5835 authors (A), 3025 papers (P), and 56 subjects (S). Paper features are the bag-of-words embedding of keywords. The papers are labeled based on their classes.

- **DBLP**. We construct a subset of DBLP by selecting 14328 papers (P) and 8789 terms (T). We further include the related 4057 authors (A) and 20 venues (V) in this subset. According to research areas,

---

**Algorithm 1** HetaFlow forward propagation.

---

**Require:** The node feature $\{\mathbf{x}_i, \forall i \in \mathcal{V}\}$,
  The number of layers $L$,
  The number of attention heads $K$,

**Ensure:** The final node embedding $\mathbf{Z}_{\mathcal{V}}$.

  **for** edge type $R \in \mathcal{R}$ **do**
    Perform BFS based flow decomposition and get a set of parallel flows $S_R = \{\mathcal{P}_1, \mathcal{P}_2, \ldots, \mathcal{P}_{n_R}\}$;

  **end for**
  Get the parallel decomposition $S$ by combine all the sets of several parallel flows $S \leftarrow \bigcup_{i=1}^{|\mathcal{R}|} S_{R_i}$;

  Calculate $T_{ee}$, $T_{ni}$, $\forall\ e \in \mathcal{E}$, $i \in \mathcal{V}$ by one hot encoding according to their types $\mathbf{T_e}_e$, $\mathbf{T_n}_i$;

  Do edge encoding $\mathbf{Z}_{\mathcal{E}e} \leftarrow T_{ni} \parallel T_{ee} \parallel T_{nj}$, $\forall\ e \in \mathcal{E}$, whereas $i$, $j$ are the end nodes of $e$;

  **for** node type $A \in \mathcal{A}$ **do**
    Node content transformation $\mathbf{h}'_i \leftarrow \mathbf{W}_A \cdot \mathbf{x}_i$, $\forall i \in \mathcal{V}_A$;

  **end for**
  **for** $l = 1 \ldots L$ **do**
    **for** parallel flow $\mathcal{P} \in$ parallel decomposition $S$ **do**
      **for** path $P \in$ parallel flow $\mathcal{P}$ **do**
        Calculate the two adaptive factors: $[\mathbf{A1}_{e_{ij}}^P]^l \leftarrow \sigma\left(\mathbf{W}_{a_1}^l \cdot \mathbf{Z}_{\mathcal{E}e_{ij}} + b_{a_1}^l\right)$, $\mathbf{A2}_P^l \leftarrow \sigma\left(\mathbf{W}_{a_2}^l \cdot T_{eP} + b_{a_2}^l\right)$;

        **for** node $i \in \mathcal{V}_P = P \cap \mathcal{V}$ **do**
          **for** node $j \in \mathcal{N}_i^p = P \cap \mathcal{N}_i$ **do**
            Calculate path-based representation $\left[\hat{\mathbf{h}}_i^P\right]^l$ using the edge-level instance encoder:

$$[\hat{\mathbf{h}}_v^P]^l \leftarrow \sum_{u \in \mathcal{N}_v^P} \sigma[w_u(\mathbf{h}'^l_u \odot \mathbf{A1}_{e_{uv}}^P)];$$

            Calculate weight coefficient $\alpha_{ij}^P$ according to (6);

          **end for**
          Combine the learned embeddings from all heads $\left[\bar{\mathbf{h}}_i^P\right]^l \leftarrow \parallel_{k=1}^K \sigma\left(\sum_{j \in \mathcal{N}_i^p} [\alpha_{ij}^P]_k \cdot \left[\hat{\mathbf{h}}_j^P\right]^l\right)$;

        **end for**
        Perform flow-based adjustments $\left[\mathbf{h}_v^P\right]^l \leftarrow \left[\bar{\mathbf{h}}_v^P\right]^l \odot \mathbf{A2}_P^l$;

        Calculate the importance weight $\beta_P$ for each path-flow;

      **end for**
      Fuse the embeddings of the same node on different paths: $\hat{\mathbf{h}}_v^l \leftarrow \Sigma_{P \in P^v} \beta_P \cdot \left[\mathbf{h}_v^P\right]^l$, $\forall\ v \in \mathcal{V}$;

    **end for**
    Update node features for the next layer with Layer output projection $[\mathbf{h}'_v]^{l+1} \leftarrow \sigma\left(\mathbf{W}_o^l \cdot \hat{\mathbf{h}}_v^l\right)$, $\forall\ v \in \mathcal{V}$;

  **end for**
  $\mathbf{Z}_{\mathcal{V}} \leftarrow \mathbf{h}_v^l$, $\forall\ v \in \mathcal{V}$;
  **return** $\mathbf{Z}_{\mathcal{V}} \forall\ v \in \mathcal{V}$;

---

Table 3: Dataset statistics.

| Datasets | Nodes | Node Types | Edges | Edge Types | Classes | Features |
|---|---|---|---|---|---|---|
| ACM | 10,942 | 4 | 547,872 | 8 | 3 | 1830 |
| DBLP | 26,128 | 4 | 239,566 | 6 | 4 | 334 |
| Freebase | 180,098 | 8 | 1,057,688 | 36 | 7 | 1 |
| IMDB | 21,420 | 4 | 86,642 | 6 | 5 | 1232 |

we place the authors into four categories: information retrieval, data mining, database, and machine learning. The research area of each author is labeled based on the conferences they participate in. We take the bag-of-words representation of keywords as the author features.

- **IMDB**. We construct a heterogeneous graph that consists of 4278 movies, 5257 actors, and 2081 directors after data preprocessing. We label the movies (Action, Comedy, and Drama) according to their genre information. Movie features are the bag-of-words representation of its plot keywords.

- **Freebase**. Freebase is a huge knowledge graph of the world's information. It involves many aspects like music, movies, people, and so on. Following the procedure of a previous survey Yang et al. (2020), we sample a subgraph of 8 categories of vertex types with about 1,000,000 edges.

Simple statistics of the datasets are summarized in Table 3, and network schema is illustrated in Appendix 8. For vertices with no attributes, we use the average of the features of their neighbors as their input features.

## 5.2 Baselines

To evaluate the performance of HetaFlow, we compare it with widely-used baselines: RGCN (Schlichtkrull et al., 2018), HAN (Wang et al., 2019), GTN (Yun et al., 2019), RSHN (Zhu et al., 2019), HetGNN (Zhang et al., 2019b), MAGNN (Fu et al., 2020), HetSANN (Hong et al., 2020), HGT (Hu et al., 2020), GCN (Kipf & Welling, 2016), GAT (Velickovic et al., 2018), MECCH (Fu & King, 2024), SeHGNN Yang et al. (2023) and Simple-HGN (Lv et al., 2021).

For comparison purposes, when testing on the homogeneous and the random-walk-based models, we first build a homogeneous subgraph, which is homogeneous in both edge and vertex types, for each meta-path. We then test the model on each subgraph. After obtaining the performance of the model for every meta-path, we take the best one as the performance of the model. For those models based on random walks, the window size is set to be 5, walk length to be 100, walks per vertex to be 40, and the number of negative samples to be 5. For a fair comparison, the embedding dimension is decided according to the best results reported in the paper Shi et al. (2018) for all models.

## 5.3 Numerical results

In all the tables, the best and second-best results for each setting are highlighted by red and blue, respectively. We consider the results to be tied for the best/second-best results if they are close (within 0.1%).

### 5.3.1 Classification results

From Table 4, we see that HetaFlow has the best performance for most cases, although it does not utilize meta-paths. In general, HetaFlow has comparable performance on the IMDB dataset and outperforms the other baselines by $0.5 - 0.8\%$ on DBLP and ACM datasets. This verifies our conjecture that parallel flow decomposition suffers less information loss than meta-path-based reconstruction methods and can make use of edge directions.

Compared to the models that simply average over vertex neighbors, GAT and Simple-HGN perform well since they train weights to weigh the information properly. Compared to HAN and MAGNN, HetaFlow, which requires no manually defined meta-paths, captures even richer semantics successfully.

Table 4: Quantitative results (%) on HGB for vertex classification task. Performances of benchmark models are cited from Yang et al. (2023), Fu & King (2024) and Lv et al. (2021).

| | ACM | | DBLP | | IMDB | |
|---|---|---|---|---|---|---|
| | Macro-F1 | Micro-F1 | Macro-F1 | Micro-F1 | Macro-F1 | Micro-F1 |
| RGCN | $91.55 \pm 0.74$ | $91.41 \pm 0.75$ | $91.52 \pm 0.50$ | $92.07 \pm 0.50$ | $58.85 \pm 0.26$ | $62.05 \pm 0.15$ |
| HAN | $90.89 \pm 0.43$ | $90.79 \pm 0.43$ | $91.67 \pm 0.49$ | $92.05 \pm 0.62$ | $57.74 \pm 0.96$ | $64.63 \pm 0.58$ |
| GTN | $91.31 \pm 0.70$ | $91.20 \pm 0.71$ | $93.52 \pm 0.55$ | $93.97 \pm 0.54$ | $60.47 \pm 0.98$ | $65.14 \pm 0.45$ |
| RSHN | $90.50 \pm 1.51$ | $90.32 \pm 1.54$ | $93.34 \pm 0.58$ | $93.81 \pm 0.55$ | $59.85 \pm 3.21$ | $64.22 \pm 1.03$ |
| HetGNN | $85.91 \pm 0.25$ | $86.05 \pm 0.25$ | $91.76 \pm 0.43$ | $92.33 \pm 0.41$ | $48.25 \pm 0.67$ | $51.16 \pm 0.65$ |
| MAGNN | $90.88 \pm 0.64$ | $90.77 \pm 0.65$ | $93.28 \pm 0.51$ | $93.76 \pm 0.45$ | $56.49 \pm 3.20$ | $64.67 \pm 1.67$ |
| HetSANN | $90.02 \pm 0.35$ | $89.91 \pm 0.37$ | $78.55 \pm 2.42$ | $80.56 \pm 1.50$ | $49.47 \pm 1.21$ | $57.68 \pm 0.44$ |
| HGT | $91.12 \pm 0.76$ | $91.00 \pm 0.76$ | $93.01 \pm 0.23$ | $93.49 \pm 0.25$ | $63.00 \pm 1.19$ | $67.20 \pm 0.57$ |
| GCN | $92.17 \pm 0.24$ | $92.12 \pm 0.23$ | $90.84 \pm 0.32$ | $91.47 \pm 0.34$ | $57.88 \pm 1.18$ | $64.82 \pm 0.64$ |
| GAT | $92.26 \pm 0.94$ | $92.19 \pm 0.93$ | $93.83 \pm 0.27$ | $93.39 \pm 0.30$ | $58.94 \pm 1.35$ | $64.86 \pm 0.43$ |
| Simple-HGN | $93.42 \pm 0.44$ | $93.35 \pm 0.45$ | $94.01 \pm 0.24$ | $94.46 \pm 0.22$ | $63.53 \pm 1.36$ | $67.36 \pm 0.57$ |
| MECCH | $92.74 \pm 0.40$ | $92.67 \pm 0.36$ | $94.34 \pm 0.29$ | $95.08 \pm 0.25$ | $62.59 \pm 1.96$ | $64.62 \pm 2.38$ |
| SeHGNN | $94.05 \pm 0.35$ | $93.98 \pm 0.36$ | $95.06 \pm 0.17$ | $95.42 \pm 0.17$ | $67.11 \pm 0.25$ | $69.17 \pm 0.38$ |
| HetaFlow | $94.42 \pm 0.33$ | $94.31 \pm 0.34$ | $95.17 \pm 0.30$ | $95.04 \pm 0.35$ | $66.79 \pm 0.28$ | $69.40 \pm 0.33$ |

Table 5: Quantitative results (%) and training time (per epoch/s) on vertex classification task with low-dimensional vertex feature. Performances of benchmark models are cited from Yang et al. (2023).

| Datasets | Metrics | RGCN | HGT | GCN | GAT | Simple-HGN | SeHGNN | HetaFlow |
|---|---|---|---|---|---|---|---|---|
| | Macro-F1 | 46.78±0.77 | 29.28±2.52 | 27.84±3.13 | 40.74±2.58 | 47.72±1.48 | 51.87 ±0.86 | 52.13±1.05 |
| Freebase | Micro-F1 | 58.33±1.57 | 60.51±1.16 | 60.23±0.92 | 65.26±0.80 | 66.29±0.45 | 65.08±0.45 | 67.10±0.57 |
| | Training time | 0.85 | 6.29 | - | - | 0.77 | 0.31 | 0.48 |

All models suffer poor performance on IMDB. This may be due to the labels of vertices: though each movie vertex may be assigned multiple labels, we merely select the most related one as its label. HetaFlow achieves a smaller performance gain on DBLP than on the other two datasets. This is mainly because it turns out that one of the manually defined meta-paths (A-P-C-P-A) contains most of the information necessary for correct classification. Thus, the performance gain is not as much as on the other two datasets ACM and IMDB. Another reason is that some vertices are discarded since they are not on the meta-paths. Therefore, GCN test only a part of the dataset and may avoid vertices that are hard to classify.

Comparing the results in Table 5 and Table 4, we observe that although SeHGNN is the best performer on IMDB, it has poor performance on Freebase. This may be due to Freebase having insufficient vertex features. We see that the performance of HetaFlow is more stable across different datasets, which suggests that the adaptive adjustments in HetaFlow are useful. The training time of GAT and GCN is not recorded since the performance is of one meta-path. HetaFlow has a fast training speed with the help of data pre-processing (the parallel flow decomposition) and lower complexity.

### 5.3.2 Clustering

We conduct the clustering task to evaluate the embeddings learned from the above algorithms. We utilize the $K$-Means algorithm to perform vertex clustering and the number of clusters $K$ is set to the number of classes for every dataset, i.e. 4 for DBLP and 3 for both IMDB and ACM. We use the same ground truth as in vertex classification. Moreover, we use the Normalized Mutual Information (NMI)(Strehl & Ghosh, 2002) and the Adjusted Rand Index (ARI)(Yeung & Ruzzo, 2001) to measure the quality of the clustering results. Since the performance of $K$-Means is highly affected by the initial centroids chosen, we repeat $K$-Means ten times for each run of the model. Furthermore, we run each model 10 times and report the averaged results in Table 6.

As seen in Table 6, on DBLP and ACM, GCN performs better than metapath2vec for the vertex classification task while metapath2vec is better for the clustering task. Random-walk-based methods (i.e., metapath2vec)

Table 6: Quantitative results (%) on vertex clustering task.

| Datasets | Metrics | metapath2vec | GCN | MAGNN | HAN | GTN | Simple-HGN | HetaFlow |
|----------|---------|--------------|-----|-------|-----|-----|------------|----------|
| ACM | NMI | 21.22±0.51 | 51.49±0.83 | 61.96±0.73 | 61.40±0.14 | 61.83±0.24 | 61.94±0.26 | 62.26±0.25 |
| | ARI | 21.00±0.41 | 53.24±0.56 | 64.79±0.27 | 64.82±0.24 | 65.40±0.34 | 65.34±0.25 | 66.05±0.24 |
| DBLP | NMI | 74.22±0.52 | 75.44±0.52 | 81.11±0.37 | 79.09±0.12 | 81.02±0.36 | 81.08±0.35 | 81.81±0.34 |
| | ARI | 78.50±0.61 | 80.91±0.97 | 84.79±0.31 | 84.56±0.28 | 85.20±0.24 | 85.08±0.35 | 85.40±0.32 |
| IMDB | NMI | 1.20±0.63 | 5.79±0.89 | 10.07±0.55 | 10.82±0.50 | 10.45±0.68 | 11.27±0.86 | 11.72±0.60 |
| | ARI | 1.70±0.46 | 3.78±1.00 | 9.78±0.55 | 10.08±0.55 | 11.53±0.53 | 11.79±0.89 | 12.40±0.62 |

Table 7: Quantitative results (%) of ablation study on parallel flow decomposition. Percentages in the header denote the sizes of training sets. 'M' means million.

| Model | Parameter Size | Score | 20% | 40% | 60% | 80% |
|-------|----------------|-------|-----|-----|-----|-----|
| HetaFlow$_{pf}$ | 4.78M | Macro-F1 | 90.72±0.29 | 91.51±0.26 | 91.97±0.23 | 92.72±0.16 |
| | | Micro-F1 | 90.76±0.30 | 91.64±0.30 | 92.28±0.19 | 92.73±0.19 |
| HetaFlow$_{wo}$ | 6.32M | Macro-F1 | 90.84±0.26 | 91.34±0.20 | 91.53±0.15 | 92.01±0.13 |
| | | Micro-F1 | 90.92±0.30 | 91.43±0.28 | 91.68±0.26 | 91.97±0.19 |
| HetaFlow$_{limit}$ | 4.98M | Macro-F1 | 89.43±0.19 | 89.97±0.14 | 90.32±0.11 | 91.26±0.11 |
| | | Micro-F1 | 89.20±0.24 | 89.61±0.18 | 90.54±0.14 | 90.92±0.08 |

have advantages in vertex clustering because the use of random walks makes vertices that have small distances in the graph to be close in the embedding space (You et al., 2019). Thus, the positional information is concerned, which facilitates the $K$-Means algorithm as it clusters vertices based on the Euclidean distances between embeddings. Despite this, HetaFlow performs consistently better than all baselines, which demonstrates that through the use of parallel flow decompositions, HetaFlow can learn a powerful vertex embedding without any meta-paths.

## 5.4 Parallel flow decomposition methods

We study two aspects of the parallel flow decomposition: the influence of introducing such decomposition and the efficiency of various decomposition methods. We test different approaches on the ACM dataset.

The main motivations for introducing parallel flow decomposition are discussed in Section 4.2. HetaFlow allows us to assign different weights to the neighboring vertices, i.e., weights are assigned based on positions instead of distances. With the same number of layers, HetaFlow models can represent a richer class of polynomials than traditional GNN approaches.

Firstly, we test the models without parallel flow decomposition nor limits on the model sizes. Secondly, we test models that employ parallel flow decomposition but with a limit on the model size. Then, we test models without parallel flow decomposition but with similar constraints on model sizes. The results are shown in Table 7

In Table 7, HetaFlow$_{pf}$ denotes our base HetaFlow model (model size is measured by the number of parameters, which is limited to 5M in this experiment), i.e., the one we used to compare with other baselines Table 6. HetaFlow$_{wo}$ is the equivalent model without utilizing parallel flow decomposition and has no limits on its model size. It decomposes the heterogeneous graph into subgraphs with homogeneous edge types instead. During convolutions, HetaFlow$_{wo}$ shall assign weights according to the distance as normal GNNs and HetaFlow$_{wo}$ also keeps the adjustment layers. HetaFlow$_{limit}$ denotes the model that does not employ parallel flow decomposition and whose model size is no larger than 5M. From Table 7, we observe that HetaFlow$_{pf}$ generally has on-par performance as HetaFlow$_{wo}$ even though the model size is much smaller. HetaFlow$_{limit}$ has the worst performance.

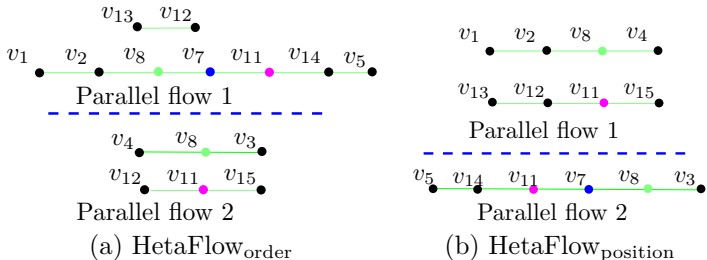

Figure 4: Examples of different parallel flow decomposition methods.

Next, we test different methods to select paths in the parallel flow decomposition using the ACM dataset, and the results are reported in Table 8. For the decomposition method mentioned in Section 4.2.1, there exist multiple ways to allocate labels to the edges of the spanning subtree.

We consider the following methods to allocate edge labels during the parallel flow decomposition procedure (all methods start from the root vertex and then target at the neighbor vertices following the descending order of their degrees in the tree):

(i) For each vertex under consideration, label adjacent edges according to the descending order of their end vertices' degrees in the original heterogeneous graph $\mathcal{G}$. For instance, consider the spanning tree whose root vertex is $v_8$ as shown in Fig. 2(a). The vertex $v_8$ has four neighbors in the tree: $v_2$ with degree 4 in the original graph, $v_3$ with degree 3, $v_4$ with degree 2, and $v_7$ with degree 3. We assign label 1 to the two edges that connect vertices with the first two highest degrees, i.e., the edge $(v_8, v_2)$ and the edge $(v_8, v_7)$. Then we remove $v_2$ and $v_7$ from the neighbor set $\mathcal{N}_v(8)$. Repeat the procedure. After all edges that connect $v_8$ are labeled, we then move to $v_2$, the neighbor vertex of $v_8$. Since one or several edges that connect our target vertex may have already been assigned with labels, we skip the corresponding labels so that no label is used more than twice. Here, we obtain the two parallel flows of Fig. 2(b) as shown in Fig. 4(a). We call this variant HetaFlow$_\text{order}$.

(ii) For each vertex $v$ under consideration, select the neighboring vertex with the highest degree in $\mathcal{G}$ as the left neighbor and the neighboring vertex with the lowest degree in $\mathcal{G}$ as the right neighbor to construct a path. Remove these from the neighbor set $\mathcal{N}_v(1)$ and repeat the procedure to obtain another path and so on. Consider the example of $v_8$ in Fig. 2(b) again. We obtain the parallel flows shown in Fig. 4(b). This variant is denoted as HetaFlow$_\text{position}$.

(iii) For each vertex under consideration, assign random labels to the edges connecting it (from 1 to $\lfloor (\deg_\text{max} + 1)/2 \rfloor$) under the constraints that no label is used more than twice. This is our base model HetaFlow$_\text{pf}$.

In addition, we also include a comparison with a clustering approach to create subcomponents as follows.

(iv) We follow Carranza et al. (2018) to construct parallel-flow-like subcomponents. This is denoted as HetaFlow$_\text{cluster}$.

From Table 8, we observe that the different parallel flow decomposition approaches yield similar performance. This suggests that HetaFlow is insensitive to the choice of parallel flow decomposition method.

## 5.5 Visualization

To visualize the results more intuitively, we project the embeddings learned by the above models into a 2-dimensional space. We visualize the author embeddings in DBLP with t-SNE (Van der Maaten & Hinton, 2008). Various colors denote the different research areas of the authors.

Table 8: Quantitative results (%) of ablation study for parallel flow decomposition method. The best and second-best results for each setting are boldfaced and underlined, respectively. Percentages in the header denote the sizes of training sets.

| Model | Score | 20% | 40% | 60% | 80% |
|---|---|---|---|---|---|
| HetaFlow$_{\text{order}}$ | Macro-F1 | 90.86±0.31 | 91.22±0.27 | 91.83±0.13 | 92.81±0.11 |
| | Micro-F1 | 90.94±0.21 | 91.46±0.15 | 92.31±0.15 | 92.87±0.11 |
| HetaFlow$_{\text{position}}$ | Macro-F1 | 90.43±0.28 | 91.67±0.27 | 91.82±0.19 | 92.56±0.14 |
| | Micro-F1 | 90.50±0.27 | 91.54±0.24 | 92.32±0.25 | 91.78±0.11 |
| HetaFlow$_{\text{pf}}$ | Macro-F1 | 90.85±0.34 | 91.43±0.31 | 92.11±0.30 | 92.78±0.28 |
| | Micro-F1 | 90.88±0.17 | 91.57±0.16 | 92.28±0.13 | 92.67±0.11 |
| HetaFlow$_{\text{cluster}}$ | Macro-F1 | 90.43±0.28 | 90.64±0.24 | 91.30±0.27 | 91.99±0.34 |
| | Micro-F1 | 90.31±0.21 | 91.04±0.18 | 91.71±0.28 | 91.58±0.20 |

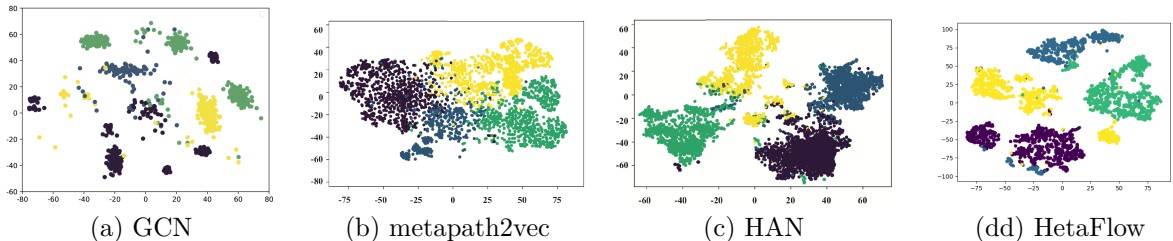

(a) GCN  (b) metapath2vec  (c) HAN  (dd) HetaFlow

Figure 5: Visualization of other embedding methods on DBLP. Each point indicates one author and its color indicates the research area.

From Fig. 5, we see that the homogeneous GNN, namely GCN, give more separate islands and the boundaries between different colors are not clear. Furthermore, we observe an area where all four types of vertices are mixed. One possible reason is that these two models only take information from one meta-path. For some vertices, their labels may be hard to tell apart under certain meta-paths.

This phenomenon also appears in the results of other models, but HAN and HetaFlow perform better as they take information from more meta-paths or the whole heterogeneous graph, respectively. For HAN, we observe that there are at most three kinds of vertices that are mixed together. Meanwhile, there exist places where all four types of vertices are mixed in the results of GCN and metapath2vec. When it comes to the result of HetaFlow, there are fewer places where more than two types of vertices are mixed. Moreover, fewer vertices, which are of different colors, are mixed when compared with the visualization of HAN.

## 6 Limitations and Future Work

In graph classification tasks that involve a multitude of graphs in both the training and test sets, HetaFlow may not perform well. This is because the parallel flow decomposition is not unique. We may get different subgraphs when performing parallel flow decomposition using different methods on the same graph. As a result, the filters and attention weights learned on one training graph may not perform well on another graph unless more restrictions are imposed to make the parallel flow decomposition unique.

In HetaFlow, the graph is decomposed into multiple parallel flows, with each vertex now appearing $\lfloor (\deg +1)/2 \rfloor$ times. Moreover, unlike meta-path-based methods, HetaFlow retains all vertices in the graph. The memory usage of HetaFlow is thus higher during runtime. A future research direction is to investigate the use of sparse matrix representations and ideas like "drop token" (Hou et al., 2022) to reduce memory storage. An optimization example is illustrated in Appendix F. Possible future research directions include the investigation of techniques like sparse matrix representations and dropping of tokens and graph edges to reduce the model

memory usage of HetaFlow. Due to its similarity to CNNs, there is also room to improve HetaFlow further by applying the wealth of methods developed for CNN models.

## 7 Conclusion

In this paper, we have proposed a new HGNN framework called HetaFlow to perform semi-supervised vertex classification and clustering tasks on heterogeneous graphs. HetaFlow addresses three limitations of existing HGNNs: loss of vertex content features due to vertex selection, the dependence on meta-path, and allocation of the same weights to neighboring vertices at the same distances. HetaFlow introduces the notion of parallel decomposition for heterogeneous graphs, which is used to enable the model to assign different weights to neighboring vertices at the same distance. This property benefits detailed vertex-level adaptive adjustments. HetaFlow applies four building block components: vertex content transformation, intra-path aggregation, adaptive adjustments, and inter-flow aggregation. In experiments, HetaFlow achieves state-of-the-art results on benchmark datasets in the vertex classification and clustering tasks.

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

# A   Proof of Proposition 1

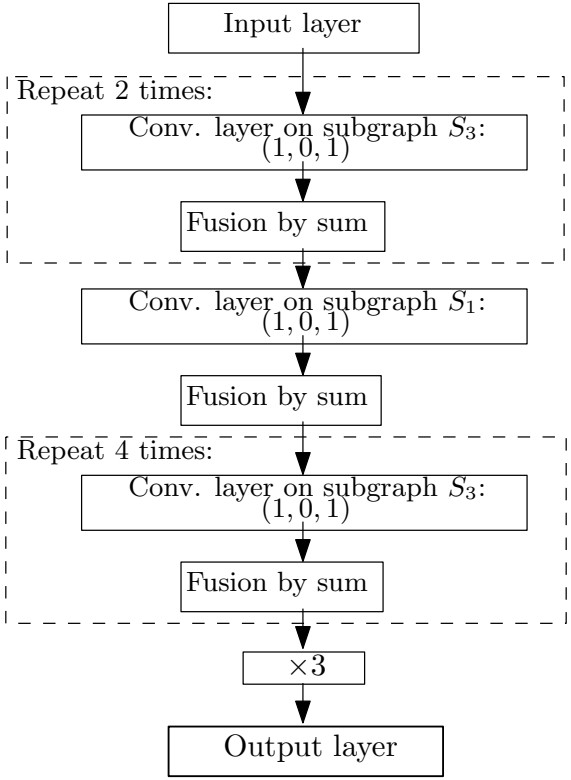

Figure 6: An example HetaFlow network to model a given monomial $3S_3^4 S_1 S_3^2$.

Given a target polynomial $\sum_{i=1}^M b_i \prod_{j=1}^N S_{a_{ij}}^{k_{ij}}$, we can construct the corresponding HetaFlow model by generating the equivalent output of each item in this polynomial and finally fuse them. The output of one monomial can be achieved by concatenating several convolution layers and fusion layers, where every convolution layer uses a 3×1 filter and fusion layers take different fusion functions with respect to the graph shift operator to be used. Here we assume that $S = A$, the adjacency matrix, so fusion layers take sum as the fusion function (same fusion function if $S$ is the Laplacian matrix; if use normalized adjacency matrix or normalized Laplacian matrix, then fusion functions should be changed to average).

Recall that $S_n$ denotes the graph shift operator of the subgraph $\mathcal{G}_n$, where $\mathcal{G}_n$ is the subgraph consisting of all parallel flows with the edge type $T_n \in \mathbb{R}^{|\mathcal{R}|}$. The proposition hypothesis assumes that there is no common edge between the parallel flows of the same subgraph. After convolutions with the filter $(1, 0, 1)$ alone each parallel flow of the subgraph $\mathcal{G}_n$, we can achieve the monomial $S_n$ with a fusion layer to sum the output of each parallel flow. For example, we can achieve the monomial $3S_3^4 S_1 S_3^2$ with the model shown in Fig. 6, where assume the graph shift to be the adjacency matrix.

For the general expression $b_i \prod_{j=1}^N S_{a_{ij}}^{k_{ij}}$, the $i$-th monomial of our target polynomial, we can choose the filter $(1, 0, 1)$ and perform convolutions on the subgraph $S_{a_{i1}}$ by $k_{i1}$ times, followed by convolutions on the subgraph $S_{a_{i2}}$ by $k_{i2}$ times and so on. HetaFlow repeats similar operations until finally obtaining $\prod_{j=1}^N S_{a_{ij}}^{k_{ij}}$. Then by multiplication with an appropriate coefficient, HetaFlow generates this monomial. The target polynomial is achieved by summing all the monomials. An explicit example of the whole model is shown in Fig. 7.

HetaFlow needs $1 + 2\sum_{j=1}^N k_{ij}$ hidden layers to generate the output of one monomial. There are thus $M + 2\sum_{i=1}^M \sum_{j=1}^N k_{ij}$ hidden layers in total in the HetaFlow model to achieve the polynomial (14).

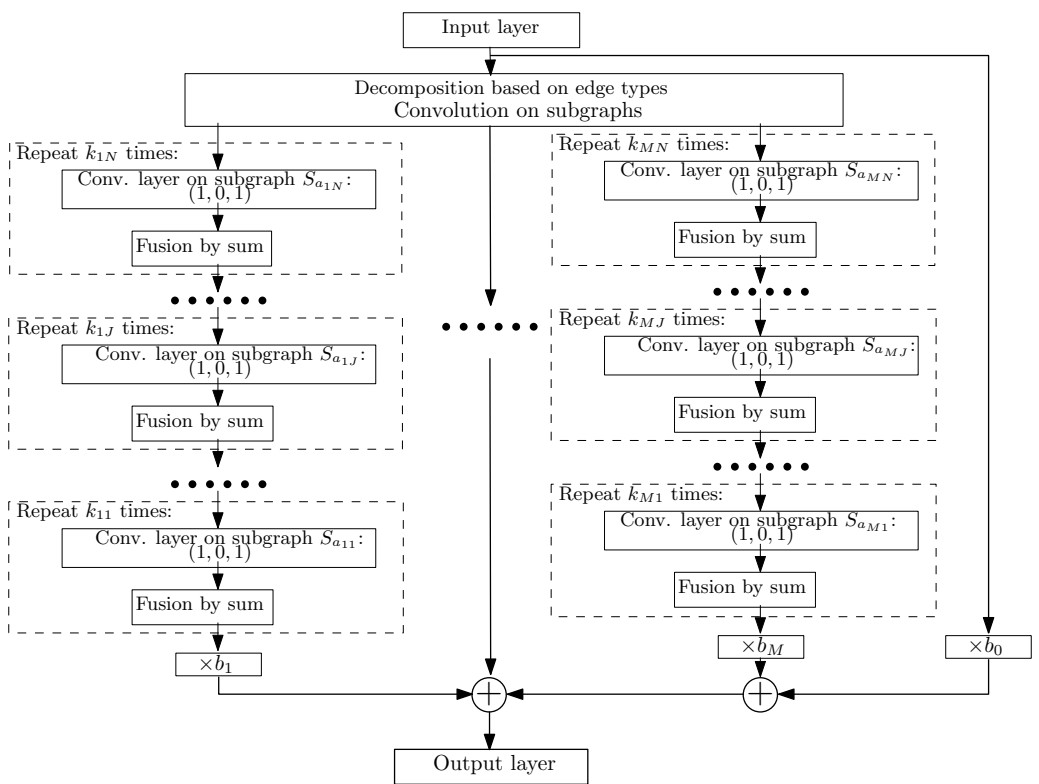

Figure 7: An example HetaFlow network to model a given polynomial.

## B Model size

According to Ji et al. (2020),

**Theorem 1.** *For any homogeneous graph $\mathcal{G}$, let $\deg_{\max}$ be the maximal degree of $\mathcal{G}$. Then*

$$\mu(\mathcal{G}) \leq (\lfloor \frac{\deg_{\max}+1}{2} \rfloor + 1) \lfloor \frac{\deg_{\max}+1}{2} \rfloor. \tag{16}$$

The above result is proved as Corollary 1 of Ji et al. (2020). We discuss the upper bound for the heterogeneous graph case in Section 4.7.

We now present an upper bound of the number of parallel flows in a parallel flow decomposition, which is a heterogeneous graph version of Theorem 1, after the following lemma.

**Lemma 1.** *Suppose $\mathcal{G} = \mathcal{G}_1 \cup \mathcal{G}_2$. Then $\mu(\mathcal{G}) \leq \mu(G_1) + \mu(G_2)$.*

*Proof.* Assume that covers of $\mathcal{G}_1$ and $\mathcal{G}_2$ are given by two sets $\mathscr{P} = \{\mathcal{P}_1, \ldots, \mathcal{P}_m\}$ and $\mathscr{Q} = \{\mathcal{Q}_1, \ldots, \mathcal{Q}_n\}$ respectively, where $\mu(G_1) = |\mathscr{P}|$ and $\mu(G_2) = |\mathscr{Q}|$. Remove all common edges of the two sets from $\mathscr{P}$ to obtain $\mathscr{R}$, where $\mathscr{R} = \mathscr{P}$ if there does not exist any common edge. Then, the union $\mathscr{R} \cup \mathscr{Q}$ is a parallel flow decomposition of $\mathcal{G}$. Thus, $\mu(\mathcal{G}) \leq |\mathscr{R}| + |\mathscr{Q}| \leq |\mathscr{P}| + |\mathscr{Q}| = \mu(G_1) + \mu(G_2)$ and the proof is complete. $\square$

**Theorem 2.** *For any heterogeneous graph $\mathcal{G}$ with $|\mathcal{R}|$ different edge types, for $i = 1, \ldots, |\mathcal{R}|$, let $\mathcal{G}_i$ be the subgraph of $\mathcal{G}$ with homogeneous edge type $i$ and $\deg_m^i$ be the maximal degree of $\mathcal{G}_i$. Then,*

$$\mu(\mathcal{G}) \leq \sum_{i=1}^{|\mathcal{R}|} \left( \lfloor \frac{\deg_m^i+1}{2} \rfloor + 1 \right) \lfloor \frac{\deg_m^i+1}{2} \rfloor. \tag{17}$$

*Proof.* By Theorem 1, for each homogeneous subgraph $\mathcal{G}_i$, $i = 1, \ldots, |\mathcal{R}|$, we have

$$\mu(\mathcal{G}_i) \leq \left( \lfloor \left( \deg_m^i + 1 \right) / 2 \rfloor + 1 \right) \lfloor \left( \deg_m^i + 1 \right) / 2 \rfloor. \tag{18}$$

Since $\mathcal{G} = \bigcup_{i=1}^{|\mathcal{R}|} \mathcal{G}_i$, the result follows from Lemma 1. $\qquad\square$

For a heterogeneous graph $\mathcal{G} = (\mathcal{V}, \mathcal{E})$ with $|\mathcal{A}|$ vertex types and $|\mathcal{R}|$ edge types, the transformation layer $t$ has $|\mathcal{A}| d_{in}^t d_{out}^t$ learnable parameters, where $d_{in}^t$ and $d_{out}^t$ are the dimensions of the input and output embeddings, respectively.

Suppose the graph $\mathcal{G}$ is decomposed into $\mu(\mathcal{G})$ parallel flows. Suppose that there are $N$ convolution layers. For convolution layer $n \in \{1, 2, \ldots, N\}$, assume that the length of the filter used is $k^n$, and the dimensions of the input and output embeddings are $d_{in}^n$ and $d_{out}^n$, respectively. Since each convolution layer employs one filter for each flow, a convolution layer $n$ contains $\mu(\mathcal{G}) k^n d_{in}^n d_{out}^n$ learnable parameters. For each convolution layer, there exists one corresponding adaptive layer and two attention parts. According to our encoding method, the dimension of edge embeddings is $2|\mathcal{A}| + |\mathcal{R}|$. To calculate the corresponding adaptive weights, the adaptive layer employs four weight matrices whose shapes are $d_{in}^n \times (2|\mathcal{A}| + |\mathcal{R}|)$, $d_{in}^n \times 1$, $d_{out}^n \times |\mathcal{A}|$ and $d_{out}^n \times 1$. The attention mechanism introduces $2(d_{in}^n + d_{out}^n)$ parameters for each convolution layer $n$.

Let $k^{\max} = \max_n k^n$, $d_{in}^{\max} = \max_n d_{in}^n$, $d_{out}^{\max} = \max_n d_{out}^n$, and $\deg^{\max} = \max_i \deg_m^i$. From Theorem 2, $\mu(\mathcal{G}) = \mathcal{O}(|\mathcal{R}|(\deg^{\max})^2)$. Then, the HetaFlow model has

$$\sum_{n \leq N} \left[ \mu(\mathcal{G}) k^n d_{in}^n d_{out}^n + d_{in}^n (2|\mathcal{A}| + |\mathcal{R}| + 3) + d_{out}^n (|\mathcal{A}| + 3) \right]$$
$$+ |\mathcal{A}| d_{in}^t d_{out}^t$$
$$\leq \mathcal{O}\Big( N|\mathcal{R}|(\deg^{\max})^2 |\mathcal{V}^{\max}| k^{\max} d_{in}^{max} d_{out}^{max}$$
$$+ N(|\mathcal{A}| + |\mathcal{R}|)(d_{in}^{max} + d_{out}^{max}) \Big) + |\mathcal{A}| d_{in}^t d_{out}^t$$

trainable parameters in total.

## C  Computational complexity

We adopt the same notations as in the previous subsection. Consider the convolution process in a parallel flow $\mathcal{P}$ in the convolution layer $n$. HetaFlow applies the convolution filter at most $|\mathcal{V}_\mathcal{P}|$ times, once at each vertex of the flow, where $|\mathcal{V}_\mathcal{P}|$ is the number of vertices contained in the parallel flow $\mathcal{P}$. For each vertex, $k^n$ dot products are computed. Thus, the computational complexity for flow $\mathcal{P}$ in the convolution layer $n$ is $\mathcal{O}(|\mathcal{V}_\mathcal{P}| k^n d_{in}^n d_{out}^n)$. Let $|\mathcal{V}^{\max}| = \max_\mathcal{P} |V_\mathcal{P}|$. The complexity for all parallel flows is bounded by $\mathcal{O}(\mu(\mathcal{G})|\mathcal{V}^{\max}| k^{\max} d_{in}^{max} d_{out}^{max})$.

Compared with the convolution process, the vertex-level adjustment process requires one extra step to calculate the weights. However, this can be omitted since the dimensions of edge-type embeddings are much smaller than the dimensions of vertex features. Following similar steps in the above discussion, the final complexity of vertex-level adjustments is $\mathcal{O}(\mu(\mathcal{G})|\mathcal{V}^{\max}| k^{\max} d_{in}^{max} d_{out}^{max})$.

As for other tasks, the complexities are $\mathcal{O}(|\mathcal{V}| d_{in}^t d_{out}^t)$ and $\mathcal{O}(|\mathcal{V}| d_{in}^p d_{out}^p)$ for transformation and prediction part respectively, where layer $t$ is the transformation layer and layer $p$ is the prediction layer. Similarly, for path-based adjustments, the complexity is $\mathcal{O}(|\mathcal{V}| d_{out}^o d_{out}^o)$ after ignoring the extra step of weight calculation, where layer $o$ is the layer that outputs features with the highest dimension.

Supposing that $d_{in}^t d_{out}^t \geq d_{in}^p d_{out}^p$ and $d_{out}^o d_{out}^o$, the total computational complexity of HetaFlow is given by

$$\mathcal{O}(\mu(\mathcal{G})|\mathcal{V}^{\max}| k^{\max} d_{in}^{max} d_{out}^{max} + |\mathcal{V}| d_{in}^t d_{out}^t) \leq \tag{19}$$
$$\mathcal{O}(|\mathcal{R}|(\deg^{\max})^2 |\mathcal{V}^{\max}| k^{\max} d_{in}^{max} d_{out}^{max} + |\mathcal{V}| d_{in}^t d_{out}^t).$$

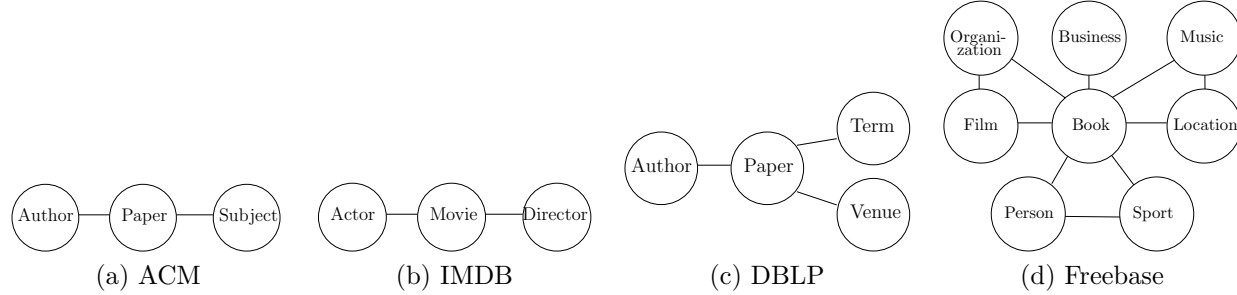

Figure 8: Network schema of the four datasets.

# D Implementation details

We employ the Adam optimizer (Kingma & Ba, 2015) to optimize the model. To find a suitable initial parameter setting, we test multiple initial hyperparameter settings, including: different dimensions of the semantic-level features $h$ from $2^6$ to $2^1$; several numbers of the attention head $K$ including $\{8, 4, 3, 2, 1\}$; different numbers of convolution layers from 1 to 4; and various dropout rates from 0.4 to 0.85 with a step size of 0.15; two learning rates including $\{0.01,\ 0.05\}$; two regularization parameters $\{0.001,\ 0.00005\}$. We compare with baselines from Lv et al. (2021) and Ahn et al. (2022). For ablation studies, we follow the settings in Wang et al. (2019). During the robust test, only part of the training set is available, namely $20\%, 40\%, 60\%, 80\%$, while the random split of training, validation, and testing sets is fixed. Furthermore, we apply early stopping with a patience of 100.

# E More ablation studies

To validate the effectiveness of each component of our model, we further conduct experiments on several variants of HetaFlow. We test the validity of vertex feature adjustment, the importance of introducing parallel flow decomposition, and the influence of different decomposition methods separately.

## E.1 Robustness test

We test the robustness of HetaFlow by limiting the size of the training set and following the settings in Wang et al. (2019). The results are shown in Appendix 9. In general, HetaFlow outperforms the other baselines by $1-3\%$ and has comparable performance when it does not. We believe this is due to parallel flow decomposition suffers less information loss than meta-path-based reconstruction methods.

## E.2 Vertex feature adjustment

We consider additive adjustments and multiplicative adjustments for comparison. For additive adjustments, we perform the following. Equation (5) for our base model uses multiplicative adjustment. For additive adjustment, we generate the type-based weight matrix $\mathbf{A}_{uv}$ in the same way as our base model (4). However, equation (5) is then changed as follows:

$$\widehat{\mathbf{h}}_v^P = \sum_{u \in \mathcal{N}_v^P} \sigma \left[ (w_u^{\mathcal{P}} + \mathbf{A}_{uv}) \mathbf{h}_u' \right]. \tag{20}$$

Furthermore, variants of multiplicative adjustment are possible. In our base model, we adjust features based on the vertex types of the target vertex and its neighboring vertex during convolution, and the edge types of their connection. See (1), (4) and (5). This method provides detailed adjustments but has high computational costs. One vertex feature may be assigned with different adjustment weights in two different convolutions on the same subgraph.

Table 9: Quantitative results (%) on vertex classification task with different training splits.

| Datasets | Metrics | Train | metapath2vec | GCN | MAGNN | HAN | GTN | Simple-HGN | HetaFlow |
|---|---|---|---|---|---|---|---|---|---|
| ACM | Macro-F1 | 20% | 64.53±0.82 | 87.20±1.04 | 87.83±0.76 | 89.92±0.23 | 90.43±0.56 | 90.59±0.75 | 91.83±0.31 |
| | | 40% | 69.08±0.77 | 87.76±0.93 | 88.48±0.62 | 90.06±0.29 | 91.12±0.49 | 91.24±0.62 | 92.18±0.37 |
| | | 60% | 70.80±0.78 | 88.14±0.97 | 90.03±0.16 | 89.15±0.26 | 92.24±0.50 | 92.08±0.54 | 92.84±0.38 |
| | | 80% | 73.97±0.73 | 90.34±0.77 | 91.87±0.76 | 90.16±0.16 | 92.81±0.67 | 92.93±0.58 | 94.10±0.31 |
| | Micro-F1 | 20% | 64.68±1.14 | 86.78±1.13 | 86.89±0.90 | 89.03±0.21 | 90.67±0.42 | 91.13±0.46 | 91.79±0.37 |
| | | 40% | 69.23±0.93 | 87.43±1.01 | 87.43±0.89 | 89.67±0.18 | 91.35±0.39 | 91.65±0.66 | 92.43±0.32 |
| | | 60% | 71.59±0.92 | 88.39±1.01 | 88.31±0.85 | 89.39±0.17 | 92.13±0.41 | 92.30±0.50 | 93.24±0.29 |
| | | 80% | 73.88±0.83 | 89.76±0.85 | 90.25±0.67 | 90.28±0.14 | 92.71±0.27 | 93.09±0.55 | 94.04±0.31 |
| DBLP | Macro-F1 | 20% | 89.91±0.88 | 89.31±0.64 | 90.56±0.97 | 90.04±0.29 | 91.13±0.42 | 91.04±0.23 | 91.48±0.34 |
| | | 40% | 89.71±0.74 | 90.81±0.68 | 92.22±0.91 | 90.92±0.28 | 92.47±0.56 | 92.53±0.28 | 92.90±0.45 |
| | | 60% | 90.46±0.78 | 90.87±0.71 | 92.79±0.76 | 91.57±0.22 | 93.73±0.61 | 93.71±0.39 | 93.43±0.42 |
| | | 80% | 90.43±0.71 | 91.25±0.66 | 93.31±0.71 | 92.16±0.22 | 93.71±0.47 | 93.91±0.20 | 94.37±0.34 |
| | Micro-F1 | 20% | 88.62±1.12 | 88.44±0.81 | 90.90±1.05 | 90.58±0.25 | 91.42±0.45 | 90.40±0.24 | 91.52±0.29 |
| | | 40% | 89.89±1.04 | 89.21±0.76 | 91.84±0.95 | 91.27±0.18 | 92.18±0.37 | 91.96±0.28 | 92.15±0.30 |
| | | 60% | 90.35±1.03 | 90.80±0.56 | 93.03±0.81 | 91.94±0.13 | 93.42±0.58 | 93.03±0.26 | 94.43±0.27 |
| | | 80% | 90.61±0.61 | 91.06±0.66 | 93.88±0.76 | 92.13±0.12 | 93.81±0.64 | 93.82±0.28 | 94.27±0.33 |
| IMDB | Macro-F1 | 20% | 40.74±1.09 | 44.85±1.15 | 49.68±1.14 | 51.10±1.19 | 51.13±1.13 | 51.59±0.98 | 52.28±1.22 |
| | | 40% | 44.47±0.76 | 47.86±1.03 | 52.11±1.01 | 52.08±1.21 | 52.43±1.21 | 53.17±1.05 | 53.89±1.20 |
| | | 60% | 44.24±0.67 | 49.07±0.71 | 53.96±0.87 | 54.08±1.18 | 54.61±1.18 | 55.44±1.26 | 56.38±1.16 |
| | | 80% | 45.56±0.70 | 51.93±0.64 | 55.29±0.74 | 55.57±1.17 | 55.40±1.16 | 56.85±1.22 | 59.42±1.08 |
| | Micro-F1 | 20% | 45.47±0.94 | 50.26±0.88 | 54.68±0.86 | 55.91±1.29 | 57.41±1.19 | 55.63±1.38 | 56.42±1.26 |
| | | 40% | 48.05±0.89 | 51.88±0.84 | 56.25±0.71 | 57.49±1.29 | 58.18±1.22 | 58.81±1.16 | 59.12±1.19 |
| | | 60% | 48.86±0.78 | 51.76±0.83 | 55.94±0.68 | 58.37±1.28 | 58.86±1.22 | 59.60±1.21 | 61.11±1.26 |
| | | 80% | 48.86±0.78 | 55.08±0.68 | 56.46±0.66 | 58.94±1.24 | 59.11±1.24 | 60.85±1.22 | 62.39±1.18 |

Another variant of multiplicative adjustment is to conduct adjustments according to the type of vertex to be adjusted and the edge types in the subgraph to be worked on, which means one vertex shares the same adjustment weights on one subgraph. For this method, the equation (1) is changed as follows:

$$\mathbf{Z}_{u\mathcal{P}} = \mathbf{T}_u \parallel \mathbf{T}_{\mathcal{P}}, \tag{21}$$

where $T_{\mathcal{P}}$ is the one-hot encoding of the edge type of parallel flow $\mathcal{P}$. This adjustment method can be performed before the convolution operation. The equations (4) and (5) are replaced by:

$$\begin{aligned}
\mathbf{A}_{u\mathcal{P}} &= \sigma\left(\mathbf{W}_a \mathbf{Z}_u + \mathbf{b}_a\right), \\
\mathbf{h}'_{u\mathcal{P}} &= \mathbf{h}'_u \odot \mathbf{A}_{u\mathcal{P}}, \\
\widehat{\mathbf{h}}^P_v &= \sum_{u \in \mathcal{N}^P_v} \sigma\left(w^{\mathcal{P}}_u \mathbf{h}'_{u\mathcal{P}}\right).
\end{aligned} \tag{22}$$

This method results in notable computational cost reduction. For instance, consider the example shown in Fig. 2. Vertex $v_{11}$ needs four adjustment weights for the base model but requires only one adjustment weight for this second approach. However, the second approach provides less detailed adjustments than our base model.

We report the results obtained from the three datasets on vertex classification in Table 10. Each score (i.e., either Macro-F1 or Micro-F1) is an average of the scores in different training proportions. Here HetaFlow$_{base}$ is our base model, i.e., the one used to compete with other baselines in Appendix 9 and Table 6. Let HetaFlow$_{wo}$ be the variant model without utilizing vertex feature adjustments; HetaFlow$_{flow}$ be the variant that follows (21) and (22), which performs flow-based multiplicative adjustments; and HetaFlow$_{add}$ be the model that uses the additive adjustment in (20). All other settings are the same for these variants of HetaFlow.

From Table 10, we observe that in most cases, adjustments used in our base model yield the best outputs, whereas flow-based multiplicative adjustment has the second-best performance. The gap between these two

Table 10: Quantitative results (%) for different adjustment methods in HetaFlow.

| Datasets | Score | wo | add | flow | base |
|---|---|---|---|---|---|
| ACM | Macro-F1 | 90.10 | 88.14 | 91.23 | 91.79 |
| | Micro-F1 | 90.31 | 88.27 | 91.44 | 91.85 |
| DBLP | Macro-F1 | 92.86 | 91.64 | 93.81 | 94.41 |
| | Micro-F1 | 93.65 | 91.99 | 94.12 | 94.72 |
| IMDB | Macro-F1 | 52.80 | 50.22 | 53.76 | 53.85 |
| | Micro-F1 | 57.69 | 55.91 | 58.39 | 58.17 |

methods is however small. The additive adjustment approach is inferior to the multiplicative adjustments. Together with the variant without vertex feature adjustment, these produce the worst performances.

## F  Drop Token Examples

Though the number of filters in HetaFlow does not scale with the size of the graph, the heterogeneity of a graph affects the number of parallel flows required. Since each parallel flow requires a separate filter, we may need to discard some edges to decrease the parallel flow number. We may need to make a trade-off between the number of parallel flows and the number of omitted edges.

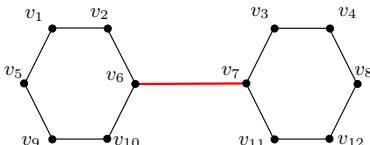

Figure 9: An example molecule graph. Different colors for edges indicate different types.

For example, consider the heterogeneous molecule graph shown in Fig. 9, which has one type of vertices and two types of edges. Assume that twelve black edges in this graph belong to edge type 1 while the red edge connecting $v_6$ and $v_7$ belong to edge type 2. To contain all edges in the parallel flow decomposition, we will need two parallel flows. One parallel flow contains the path $P_1 = (v_6, v_7)$ while the other consists of two paths $P_2 = (v_1, v_2, v_6, v_{10}, v_9, v_5, v_1)$ and $P_3 = (v_3, v_4, v_8, v_{12}, v_{11}, v_7, v_3)$. However, if we discard the type 2 edge, then we only need one parallel flow. This means that by discarding the edge $(v_6, v_7)$, we reduce the size of our model almost by half while only losing 7.7% of the edges. Research on methods that choose the edges to be discarded is part of our future work.

HetaFlow's convolution filter assigns various weights to the neighbor vertices based on the position relationships between them and the central vertex. On the other hand, if one chooses the graph adjacency or Laplacian matrix as the graph shift operator for a GCN, then a common weight is shared by the neighbors with the same distance to the central vertex. In this regard, HetaFlow is much closer to CNN, which means it may benefit from techniques developed for CNNs. Our future work includes investigations into this observation.

