# OpenReview forum: "Heterogeneous graph adaptive flow network"
_TMLR — Accepted by TMLR_

### Review · Reviewer_TKc7 · 2024-05-28

**Summary Of Contributions:**

This work proposes Heterogeneous graph adaptive flow network (HetaFlow), which removes the need for meta-paths. HetaFlow decomposes the heterogeneous graph into flows and performs convolution across heterogeneous vertex and edge types, using an adaptation to change the vertex features based on the corresponding vertex and edge types during aggregation. Authors perform experiments for vertex clustering and vertex classification and release their source code for reproducibility.

**Audience:**

Yes

**Claims And Evidence:**

Yes

**Requested Changes:**

Please see the above section.

**Strengths And Weaknesses:**

Figure 1 is informative in describing the problem setting of heterogenous graphs and Introduction section summarizes the research work’s contributions comprehensively.

This work is missing references to an important state-of-the-art GNN model that outperforms HAN without using metapaths. This work should reference that model in the “Related Work” section for recency and relevance:
"Bi-Level Attention Graph Neural Networks," in 2021 IEEE International Conference on Data Mining (ICDM), Auckland, New Zealand, 2021 pp. 1126-1131. doi: 10.1109/ICDM51629.2021.00133

Section 3 is well-written and useful for understanding background on notation.

In Section 4, the authors describe the components of their model architecture, however novelty of the work seems to be limited. Specifically, the attention mechanism used seems to be highly similar to GAT based attention and Transformer based attention. Works like HAN already learn attention at both node and edge (relation) level. The authors need to more clearly explain what the novelty of their work is compared to the existing state-of-the-art.

I also have a concern that the size of the dataset being evaluated is small-scale (limited to thousands of nodes), when there are several large-scale datasets (RDF) that contain millions of nodes. Can the authors also evaluate on these datasets and provide a section on the memory and runtime complexity of the model?

Experiment results achieve good performance and the authors have identified limitations and future directions.

---

> ### Author Response · Authors · 2024-06-26
> **Response to Reviewer TKc7**
>
> >Comment #1:
> Figure 1 is informative in describing the problem setting of heterogenous graphs and Introduction section summarizes the research work’s contributions comprehensively.
>
> Response #1:
> Many thanks for the reviewer’s comments.
>
> >Comment #2:
> This work is missing references to an important state-of-the-art GNN model that outperforms HAN without using metapaths. This work should reference that model in the “Related Work” section for recency and relevance: "Bi-Level Attention Graph Neural Networks," in 2021 IEEE International Conference on Data Mining (ICDM), Auckland, New Zealand, 2021 pp. 1126-1131. doi: 10.1109/ICDM51629.2021.00133
>
> Response #2:
> We apologize for our negligence in missing these references and many thanks for your reminder. We have added it and some other references, e.g., [1,2,3]. According to the reviewer’s suggestion, we have rewritten the introduction part and discussed the models in those newly added references at the end of Section 2.
> Refs:
> [1] [WWW 2020] [HGT] Heterogeneous Graph Transformer
> [2] [AAAI 2020] [HetSANN] An Attention-based Graph Neural Network for Heterogeneous Structural Learning
> [3] [TKDE 2021] [ie-HGCN] Interpretable and Efficient Heterogeneous Graph Convolutional Network
>
> >Comment #3: In Section 4, the authors describe the components of their model architecture, however novelty of the work seems to be limited. Specifically, the attention mechanism used seems to be highly similar to GAT based attention and Transformer based attention. Works like HAN already learn attention at both node and edge (relation) level. The authors need to more clearly explain what the novelty of their work is compared to the existing state-of-the-art.
>
> Response #3:
> We thank the reviewer for this valuable suggestion. We have added a new Section 4.1 to explain the novelty more clearly and moved the old Section 4.1.1 into Section 4.7. The novelty of our model is that it learns filters that assign weights according to position instead of distance. Most existing methods failed to discuss the influence of ‘path directions’ in heterogeneous graphs. For instance, assume there exists a meta path ‘paper-author-conference-paper’ in the ACM dataset. Then the graph generated based on this meta path is a directed graph, which means the neighbor vertices of the same distance are likely to have different importance to this center vertex. By decomposing the graph into 1-D flows, we enable the model to learn different weights to 1-hop neighbors. Thus, this model can do aggregations more efficiently.
>
> We have added concrete aggregation examples and compared HetaFlow with other methods (both meta-path-based and non-meta-path-based methods) at the beginning of Section 4. We also moved the theoretic result part (Section 4.1.1 previously) to Section 4.7.1 (after the method was claimed). This part shows the good expressiveness of HetaFlow.
>
> >Comment #4: I also have a concern that the size of the dataset being evaluated is small-scale (limited to thousands of nodes), when there are several large-scale datasets (RDF) that contain millions of nodes. Can the authors also evaluate on these datasets and provide a section on the memory and runtime complexity of the model?
>
> We have done more experiments on a large dataset Freebase according to the reviewer’s suggestion. The results are shown as follows. We have added the results and corresponding discussions in Section 5. The best and second-best results for each setting are boldfaced and italicized, respectively.
>
> | Freebase | RGCN           | HGT            | GCN            | GAT            | Simple-HGN     | SeHGNN         | HetaFlow       |
> |----------|----------------|----------------|----------------|----------------|----------------|----------------|----------------|
> | Macro-F1 | 46.78$\pm$0.77 | 29.28$\pm$2.52 | 27.84$\pm$3.13 | 40.74$\pm$2.58 | 47.72$\pm$1.48 | *51.87$\pm$0.86* | **52.13$\pm$1.05** |
> | Micro-F1 | 58.33$\pm$1.57 | 60.51$\pm$1.16 | 60.23$\pm$0.92 | 65.26$\pm$0.80 | *66.29$\pm$0.45* | 65.08$\pm$0.45 | **67.10$\pm$0.57** |
>
> We added a section 4.7.2 to compare the time complexity of HetaFlow against other models (HGT, HAN, and SeHGNN). Results are summarized in the following table. The complexity of HetaFlow is lower when trying to consider all possible meta-paths in complicated graphs. Detailed theoretical analysis of model size and computational complexity of HetaFlow are provided in Appendix B and C. We just finished the tests for HetaFlow(0.38s/epoch), HAN (11.52s/epoch), and Simple-HGN (0.77s/epoch).
>
> |       |  SeHGNN    |  HAN      |  Simple-HGN  |  HetaFlow        |
> |-------|----------------------|------------------------|-----------------------|----------------------------------------|
> | Total | $O(nd(M^2{+}Md))$ | $O(nd(Me_1{+}Md))$ | $O(nd(e_2{+}ld))$ | $O( (\|R'\|\|P'\|+l)nd^2)$ |

---

> > ### Comment · Reviewer_TKc7 · 2024-06-30
> > **Review Acknowledgement**
> >
> > Thanks for your response. I have read these responses and have made my recommendation accordingly.

---

> > > ### Author Response · Authors · 2024-07-01
> > > **Thank you for spending your precious time to review our paper**
> > >
> > > Thank you for spending your precious time to review our paper. Your comments and suggestions are very helpful for us to improve the work.

---

### Review · Reviewer_QmFN · 2024-06-04

**Summary Of Contributions:**

This submission contributes a new heterogeneous graph neural network (HGNN) method to address the limitation that previous HGNN methods require meta-paths.

**Audience:**

Yes

**Broader Impact Concerns:**

I did not identify any concerns on the ethical implications of the work.

**Claims And Evidence:**

Yes

**Requested Changes:**

The authors are suggested to enhance the motivation by investigating more existing HGNN methods that do not require meta-paths [1,2,3], especially the work that can automatically search meta-paths [3]. This is critical to securing my recommendation.

**Strengths And Weaknesses:**

Strengths

- The authors provide the codes of the work, facilitating good reproducibility.
- The experiments are extensive and the results are promising.
- The concept of parallel flow defined in this work is interesting.

Weaknesses

- The major concern of this work is that the motivation of this work is fragile. There exist HGNN methods that do not require meta-paths, e.g., [1,2,3]. In particular, the previous HGNN method [3] has been proven to be able to automatically discover and exploit useful meta-paths from all possible ones. These works should be discussed in the text, and the difference and the connection between them and the proposed work should be analyzed.
- I am confused about why propose the new local structure of parallel flow.
- The sentence below Definition 3 seems to be incomplete.


Refs:

[1] [WWW 2020] [HGT] Heterogeneous Graph Transformer

[2] [AAAI 2020] [HetSANN] An Attention-based Graph Neural Network for Heterogeneous Structural Learning

[3] [TKDE 2021] [ie-HGCN] Interpretable and Efficient Heterogeneous Graph Convolutional Network

---

> ### Author Response · Authors · 2024-06-26
> **Response to Reviewer QmFN**
>
> >Comment #1:Strengths
> •	The authors provide the codes of the work, facilitating good reproducibility.
> •	The experiments are extensive and the results are promising.
> •	The concept of parallel flow defined in this work is interesting.
>
> Response #1: Many thanks for the reviewer’s comments.
>
> >Comment #2:Weaknesses
> •	The major concern of this work is that the motivation of this work is fragile. There exist HGNN methods that do not require meta-paths, e.g., [1,2,3]. In particular, the previous HGNN method [3] has been proven to be able to automatically discover and exploit useful meta-paths from all possible ones. These works should be discussed in the text, and the difference and the connection between them and the proposed work should be analyzed.
>
> Response #2: We have rewritten the introduction part according to the reviewer’s suggestion. We have added the models mentioned and some other references as well, namely [1,2]. We did more comparisons against non-meta-path-based methods in the introduction part. We also analyzed the differences and connections between our model and them.
> In order to explain our motivation more clearly, we have also added concrete aggregation examples and compared HetaFlow with other methods at the end of Section 2 and in Section 4.1. We claimed that our model was better at making use of the directions of edges and learned more suitable representations.
> Refs:
> [1] [ICDM 2021] [BA-GNN] Bi-Level Attention Graph Neural Networks
> [2] [AAAI 2023] [SeHGNN] Simple And Efficient Heterogeneous Graph Neural Network.
>
> >Comment #3:•	I am confused about why propose the new local structure of parallel flow.
>
> Response #3: We apologize for not presenting our motivation for employing parallel flow clearly. We have added a new Section 4.1 to discuss the novelty/motivation.
> The novelty of introducing parallel flow is that it enables the model to learn aggregation weights according to positions instead of distances. Most existing methods failed to consider the influence of ‘path directions’ in heterogeneous graphs.
> For instance, assume there exists a meta path ‘paper-author-conference-paper’ in the ACM dataset. Then the graph generated based on this meta path is a directed graph, which means the neighbor vertices of the same distance are likely to have different importance to this center vertex. Many current HGNNs generate ‘new graphs’ so that they can do the aggregation step, but they ignore the edge directions of these ‘new graphs’. Please refer to Section 4.1. for more details.
>
> >Comment #4:•	The sentence below Definition 3 seems to be incomplete.
>
> Response #4: Many thanks for the reviewer’s reminder. We have rewritten this sentence according to the reviewer’s suggestion.
>
> >Comment #5:Requested Changes:
> The authors are suggested to enhance the motivation by investigating more existing HGNN methods that do not require meta-paths [1,2,3], especially the work that can automatically search meta-paths [3]. This is critical to securing my recommendation.
>
> We thank the reviewer for this valuable suggestion. We have rewritten the middle of Section 1 and the end of Section 2 according to the reviewer’s suggestion. We have added the comparisons against models that do not require meta-paths at the end of Section 2. We also added concrete aggregation examples at the beginning of Section 4 to claim our motivation to employ parallel flow structures. Please see above about the novelty of introducing parallel flow.

---

### Review · Reviewer_dAfN · 2024-06-06

**Summary Of Contributions:**

The authors study learning on heterogeneous graphs consisting of varied types of nodes. To deal with heterogeneity, the authors create a decomposition of the graph into paths following a particular edge type (called a parallel flow decomposition). Then, the model (HetaFlow) implements a convolution and attention within each path, and then between the different paths.

**Audience:**

Yes

**Claims And Evidence:**

No

**Requested Changes:**

Major
- (Critical) The introduction spends a significant amount of time comparing the method to meta-path-based methods, but this comparison is not made as clearly in the paper. I think there is a major theoretic result in 4.1.1, but this was difficult to follow because it was introduced before the method was introduced. Furthermore, it does not directly relate to the points made in the introduction, leaving the reader to infer what exactly this proposition implies. It should be moved until after
- (Critical) Similarly, the paper claims that meta-path-based methods are not scalable and that Hetaflow addresses this challnge, but no comparison on scalability is performed empirically. The authors claim in the limitations that the memory use of HetaFlow is higher (which makes sense, as it has separate weights for each parallel flow). Although future directions for scalability are proposed, this claim could use more support by showing runtime and memory usage for both, and showing how many parallel flows and paths are used in each dataset.
- (Strengthen) Additionally, scalability is further shown by using a dataset that is an order of a magnitude larger and comparing runtimes/memory usage.
- (Critical) The related work compares HetaFlow to Simple-HGN by saying that it has over-smoothing and limited receptive field. However, it is not argued why HetaFlow overcomes these issues. In general, I'm surprised why meta-path-based methods are singled out in the introduction, even though Simple-HGN seems to be performing best among the baselines, and further comparison could help convince the reader.
- (Critical) Section 4 could use a restructure. Start with a high level overview of the method before diving into details. Section 4.1.1 should come after the method is introduced: The proposition mentions 'HetaFlow model', but this concept is not yet defined. This section also needs more high-level explanation: Why is a polynomial better? And isn't $M+2\sum_i^ M\sum_j^ N k_{ij}$ a massive number of layers, suggesting this result is not as powerful?
- (Strengthen) A pseudoalgorithm figure could greatly help to understand the total computation.


Minor
- (Critical) It is unclear if the authors assume there is at most 1 edge between each pair of nodes
- (Critical) Section 4.1.2: It is not clear how edges in the tree are labelled with a number. I don't think there's a unique way to do this, so it should be mentioned.
- (Critical) Section 4.3: It wasn't clear to me if there's a weight $w_u^ P$ for every node, or if it is just unique to each parallel flow.
- (Critical) What is the activation function $\sigma$?
- (Critical) Motivate why intra-flow aggregation uses 'Bahdanau attention' with concatenation while inter-flow aggregation uses the (currently more standard) dot-product attention.

**Strengths And Weaknesses:**

Strengths: I like the ideas behind the approach, although I do not have the expertise to judge the novelty. The design of the model seems decent, and the experiments are also decent, showing significant performance increases on the baselines (although with increased complexity).

Weaknesses: The explanation of the method is fairly difficult to follow and quite long. It could use a restructuring. The paper compares in the introduction to meta-path-based methods, but the arguments made here are not convincingly supported by evidence in the paper. The paper primarily compares to heterogeneous GNN methods from before 2020, if I understand correctly. I do not have the expertise to judge if these baselines are strong (enough).

---

> ### Author Response · Authors · 2024-06-26
> **Response to Reviewer dAfN (PART 1/3)**
>
> >Comment #1: Strengths: I like the ideas behind the approach, although I do not have the expertise to judge the novelty. The design of the model seems decent, and the experiments are also decent, showing significant performance increases on the baselines (although with increased complexity).
>
> Response #1: Many thanks for the reviewer’s comments.
>
> >Comment #2: Weaknesses: The explanation of the method is fairly difficult to follow and quite long. It could use a restructuring. The paper compares in the introduction to meta-path-based methods, but the arguments made here are not convincingly supported by evidence in the paper. The paper primarily compares to heterogeneous GNN methods from before 2020, if I understand correctly. I do not have the expertise to judge if these baselines are strong (enough).
>
> Response #2:  We restructured the explanation of the method to make it clearer.
> We have rewritten the introduction part according to the reviewer’s comments. We have investigated more existing HGNN methods that do not require meta-paths [1,2,3,4]. We did more comparisons against non-meta-path-based methods at the end of Section 2. We also analyzed the differences and connections between our model and them.
> To make the baselines strong enough, we added comparisons against SeHGNN (2023) [5], MECCH (first submitted in 2022 and lately revised in Nov 2023).
> Refs:
> [1] [WWW 2020] [HGT] Heterogeneous Graph Transformer
> [2] [AAAI 2020] [HetSANN] An Attention-based Graph Neural Network for Heterogeneous Structural Learning
> [3] [TKDE 2021] [ie-HGCN] Interpretable and Efficient Heterogeneous Graph Convolutional Network
> [4] [ICDM 2021] [BA-GNN] Bi-Level Attention Graph Neural Networks
> [5] [AAAI 2023] [SeHGNN] Simple And Efficient Heterogeneous Graph Neural Network
> [6] [WWW 2024] [AdaptKry] Optimizing Polynomial Graph Filters: A Novel Adaptive Krylov Subspace Approach
>
> >Comment #3: Requested Changes:
> Major
> •	(Critical) The introduction spends a significant amount of time comparing the method to meta-path-based methods, but this comparison is not made as clearly in the paper. I think there is a major theoretic result in 4.1.1, but this was difficult to follow because it was introduced before the method was introduced. Furthermore, it does not directly relate to the points made in the introduction, leaving the reader to infer what exactly this proposition implies. It should be moved until after
>
> Response #3: We have rewritten the introduction and related works part according to the reviewer’s comments. We investigated more existing HGNN methods that do not require meta-paths [1,2,3,4]. We also added a new Section 4.1 to explain the novelty more clearly. Section 4.1 contains some concrete aggregation examples to make the explanation clearer. These examples also help us to further analyze the differences and connections between HetaFlow and those models.
> The proposition mainly claims the generality of our model. We have moved the major theoretical result in Section 4.1.1 to Section 4.7.
>
> >Comment #4: •	(Critical) Similarly, the paper claims that meta-path-based methods are not scalable and that Hetaflow addresses this challnge, but no comparison on scalability is performed empirically. The authors claim in the limitations that the memory use of HetaFlow is higher (which makes sense, as it has separate weights for each parallel flow). Although future directions for scalability are proposed, this claim could use more support by showing runtime and memory usage for both, and showing how many parallel flows and paths are used in each dataset.
>
> Response #4: We thank the reviewer for this valuable suggestion.
> By ‘not scalable’, we mean that if the ‘meta-path-based approaches’ do not discard any meta-paths, they will need a large number of filters. For HetaFlow, it just needs to generate ‘subgraphs with the same edge type’ for each edge type and then perform the flow decomposition. Typically, each subgraph has less than 10 parallel flows. So, HetaFlow needs much fewer filters when the number of vertex types is large.
> We illustrate with a concrete example. Assume that the meta-path-based approach considers all possible length 5 meta-paths (we assume the endpoints of meta-paths are of the same node type). When there are 30 node types, each position on the meta-path has 30 possibilities. There are in total $30^4=810000$ possible meta-paths. So meta-path-based model shall need 810000 filters. Similarly, there are at most $30^2=900$ edge types. So HetaFlow needs to do at most 900 flow decompositions. Typically, each subgraph has less than 10 parallel flows. So, HetaFlow needs no more than 9000 filters.
> We added Section 4.7.2 to discuss the time complexity which has a similar analysis. Detailed analysis of the model size and computational complexity is in Appendix B and C.

---

> ### Author Response · Authors · 2024-06-26
> **Response to Reviewer dAfN (PART 2/3)**
>
> >Comment #5: •	(Strengthen) Additionally, scalability is further shown by using a dataset that is an order of a magnitude larger and comparing runtimes/memory usage.
>
> Response #5: We have added tests of HetaFlow on the large dataset Freebase and the results are as follows. The results of other models are cited from [5].
>
> | Freebase | RGCN   | HGT    | GCN    | GAT    | Simple-HGN   | SeHGNN     | HetaFlow   |
> |---|---|---|---|---|---|---|---|
> | Macro-F1 | 46.78$\pm$0.77 | 29.28$\pm$2.52 | 27.84$\pm$3.13 | 40.74$\pm$2.58 | 47.72$\pm$1.48 | *51.87$\pm$0.86* | **52.13$\pm$1.05** |
> | Micro-F1 | 58.33$\pm$1.57 | 60.51$\pm$1.16 | 60.23$\pm$0.92 | 65.26$\pm$0.80 | *66.29$\pm$0.45* | 65.08$\pm$0.45 | **67.10$\pm$0.57** |
>
> We added a section 4.7.2 to compare the time complexity of HetaFlow against other models. Results are summarized in the following table. The complexity of HetaFlow is lower when trying to consider all possible meta-paths in complicated graphs. Detailed theoretical analysis of model size and computational complexity of HetaFlow are provided in Appendix B and C. We just finished the tests for HetaFlow(0.38s/epoch), HAN (11.52s/epoch), and Simple-HGN (0.77s/epoch).
>
> |       |  SeHGNN    |  HAN      |  Simple-HGN  |  HetaFlow        |
> |---|---|----|----|---|
> | Total | $O(nd(M^2{+}Md))$ | $O(nd(Me_1{+}Md))$ | $O(nd(e_2{+}ld))$ | $O( (\|R'\|\|P'\|+l)nd^2)$ |
>
> >Comment #6: •	(Critical) The related work compares HetaFlow to Simple-HGN by saying that it has over-smoothing and limited receptive field. However, it is not argued why HetaFlow overcomes these issues. In general, I'm surprised why meta-path-based methods are singled out in the introduction, even though Simple-HGN seems to be performing best among the baselines, and further comparison could help convince the reader.
>
> Response #6: We have rewritten and reclarified ourselves about this part.
> We have investigated more existing HGNN methods that do not require meta-paths [1,2,3,4]. To explain our motivation more clearly, we added a new section 4.1 to discuss the novelty and our motivation. We did more comparisons against non-meta-path-based methods at the end of Section 2 and in Section 4.1. We also analyzed the differences and connections between our model and them.
>
> >Comment #7:•	(Critical) Section 4 could use a restructure. Start with a high level overview of the method before diving into details. Section 4.1.1 should come after the method is introduced: The proposition mentions 'HetaFlow model', but this concept is not yet defined. This section also needs more high-level explanation: Why is a polynomial better? And isn't $\mathbf{w}_k \mathbf{P}^k M+2\sum_i^ M\sum_j^ N k_{ij}$ a massive number of layers, suggesting this result is not as powerful?
>
> Response #7: We have rewritten this part according to the Reviewer’s suggestion. We have moved the theoretical result part to Section 4.6 and added some concrete examples, comparisons, and analysis in Section 4.1.
> Polynomial graph filters have been widely used as guiding principles in the design of Graph Neural Networks (GCN, GAT, Simple-HGN, and so on). Huang et al. 2024 [6] proved that polynomial graph filters can be simply expressed in a uniform using Krylov basis as  $\mathbf{z}=\sum_{k=0}^K \mathbf{w_k} \mathbf{P}^k \cdot \mathbf{x}$.
> The polynomial is a general expression of these GCN models. Our Proposition 1 claims that HetaFlow can achieve the same output as the convolution filters for any graph input signal. We want to use this proposition to prove the generality of HetaFlow.
> $M+2\sum_i^ M\sum_j^ N k_{ij}$ is indeed a massive number of layers. But most other methods (if can achieve the polynomial) need a similar number of layers to achieve it as well. For example, Simple-HGN also needs $1+\sum_j^ N k_{ij}$ hidden layers to generate the output of one monomial. it requires employing M Simple-HGNs to achieve the polynomial, which means a similar number of layers is required.
> HetaFlow can achieve the same output as Simple-HGN, but simple models like Simple-HGN cannot generate the same output when HetaFlow employs unsymmetric filter weights. Thus, HetaFlow is more general.
>
> >Comment #8:•	(Strengthen) A pseudoalgorithm figure could greatly help to understand the total computation.
>
> Response #8: We thank the reviewer for this valuable suggestion and added one pseudo algorithm figure at the beginning of Section 4 (shown on page 12).
>
> >Comment #9:Minor
> •	(Critical) It is unclear if the authors assume there is at most 1 edge between each pair of nodes
>
> Response #9: Many thanks for asking. We do not have such an assumption. We view multiple edges as different edges. Consider the example in Figure 2. If there are two green edges between $v_8$ and $v_7$, and three green edges between $v_7$ and $v_{11}$, then we can still do the flow decomposition. Figure 2.(d) shall consist of 4 parallel flows, whereas parallel flow 3 is ‘$v_8$-$v_7$-$v_{11}$’ and parallel flow 4 is $v_7$-$v_{11}$.

---

> ### Author Response · Authors · 2024-06-26
> **Response to Reviewer dAfN (PART 3/3)**
>
> >Comment #10:•	(Critical) Section 4.1.2: It is not clear how edges in the tree are labelled with a number. I don't think there's a unique way to do this, so it should be mentioned.
>
> Response #10: This is a very good question. There indeed exist many ways to label the edges. In this part, however, we are just trying to provide a brief general guide. We want to show that we can always do flow decomposition to heterogeneous graphs.
>
> We have performed ablation studies about the influence of different decomposition rules, e.g., label them following the descending order of their degrees in the tree, and label them randomly. We also included a comparison with a clustering approach to create subcomponents. We tested the performance of these variants but just found that they are neck to neck. Since the performances are close, we finally used the random label approach in implementation (because it is the most simple way).  We added ablation studies about the influence of different decomposition rules in Section 5.4
>
>
> >Comment #11:•	(Critical) Section 4.3: It wasn't clear to me if there's a weight $w_u^ P$ for every node, or if it is just unique to each parallel flow.
>
> Response #11: it is unique to each parallel flow.
> There may be several sets of parallel flows. This is because HetaFlow can generate ‘a subgraph with homogeneous edge type’ for each edge type/path. So, one heterogeneous graph may have several ‘subgraphs’ to be decomposed and each subgraph decomposition shall give a set of parallel flows. That’s why there are two attention mechanisms after aggregation. One for the feature fusion inside each set of parallel flows and one for the fusion among sets.
> We illustrate with a concrete example. Assume that there are 6 edge types in total. Then HetaFlow can generate 6 ‘subgraphs with homogeneous edge type’ and perform flow decomposition on them. Assume that each subgraph is decomposed into 4 parallel flows. Then Heta flow would need $4 \times 6 =24$ filters in total.
>
> >Comment #12:•	(Critical) What is the activation function $\sigma$?
>
> Response #12: After some tests, we just used leaky ReLU.
>
> >Comment #13:•	(Critical) Motivate why intra-flow aggregation uses 'Bahdanau attention' with concatenation while inter-flow aggregation uses the (currently more standard) dot-product attention.
>
> Response #13: This is a very good question. The intra-flow aggregation involves adjustment weights (eq. (6)), which is closely related to the encoding of the type of edges. If the initial value of some parameters, like $W_a$ in eq. (6), is bad, then sometimes the learning process may become very slow at the beginning. We found that by introducing concatenation, we can speed up the learning process and keep the performance.

---

> > ### Comment · Reviewer_dAfN · 2024-07-04
> >
> > I thank the authors for their extensive response, which cleared up several of my issues. I took the response into account for my final recommendation.

---

### Decision · Action_Editor_hxys · 2024-09-13

**Recommendation:** Accept as is

**Comment:**

The paper addresses the task of vertex clustering and classification in heterogenous graphs. It proposes a decomposition of heterogeneous graphs into a set of paths called flows. Each flow contains homogeneous edges and potentially heterogeneous vertices. These are processed in parallel, aggregating information within each path, and between paths.

The approach is computationally efficient, albeit at a higher memory cost. Its performance on benchmark datasets meets or exceeds meta-path based approaches and other approaches.

The reviewers had some concerns around the structure of the paper, missed references, comparison to non-meta-path-based approaches, additional complexity analysis and comparison, lack of large-scale benchmarks, and distinguishing the novelty of the approach from GAT networks. These were all addressed in the revision. There was a minor comment from one of the reviewers about oversmoothing, but that is orthogonal to this work and not mentioned as one of the claims in the paper.

Minor comments:
- One minor request is that the bold numbers in the tables only correspond to the highest mean, whereas if two numbers are within standard error then I think that they should both be bolded.

- The responses on OpenReview include timing information (per epoch), but these are absent from the current version of the paper. I think that including these would be informative.

**Audience:**

Yes, researchers interested in heterogeneous graph neural networks would likely be interested in this paper.

**Claims And Evidence:**

The main claims are:
- Overcoming the limitations of meta-path algorithms
- Reduced computational complexity compared to meta-path approaches
- Better information preservation and exploitation (preserving directionality of edges, information loss from path selection, using type information, etc.)
- Superior performance on vertex clustering and classification

These paper in its current form satisfies these claims.

---

> ### Author Response · Authors · 2024-10-18
> **Many thanks for your precious time and constructive feedback.**
>
> We would like to thank all reviewers and AE for their helpful comments and suggestions. We have thoroughly revised the paper based on the discussions.